# TAMING AI BOTS: CONTROLLABILITY OF NEURAL STATES IN LARGE LANGUAGE MODELS

## ABSTRACT

We tackle the question of whether an agent can, by suitable choice of prompts, control an AI bot to any state. We view large language models (LLMs) and their corresponding conversational interfaces (AI bots) as discrete-time dynamical systems evolving in the embedding space of (sub-)word tokens, where they are trivially controllable. However, we are not interested in controlling AI Bots to produce individual words but rather sequences, or sentences, that convey certain "meanings". To tackle the question of controllability in the space of meanings, we first describe how meanings are represented in an LLM: after pre-training, the LLM is a deterministic map from incomplete sequences of discrete tokens to an inner product space of discriminant vectors ("embeddings") of the next token; after fine-tuning and reinforcement, the same LLM maps complete sequences to a vector space. Since no token follows the special end-of-sequence token during pre-training, that vector space can be co-opted to represent meanings and align them with human supervision during fine-tuning. Accordingly, "meanings" in trained LLMs can be viewed simply as equivalence classes of complete trajectories of tokens. Although rudimentary, this characterization of meanings is compatible with so-called deflationary theories in epistemology. More importantly, defining meanings as equivalence classes of sentences allows us to frame the key question as determining the controllability of a dynamical system evolving in the quotient space of discrete trajectories induced by the model itself, a problem that to the best of our knowledge has never been tackled before. To do so, we characterize a "well trained LLM" through conditions that are largely met by today's LLMs and show that, when restricted to the space of meanings, a well-trained AI bot is controllable under verifiable conditions. More precisely, we introduce a functional characterization of AI bots, and derive necessary and sufficient conditions for controllability. The fact that AI bots are controllable means that they can be designed to counteract adverse actions and avoid reaching undesirable states before their boundary is crossed.

## 1 INTRODUCTION

In early 2023, public access to a popular AI bot was limited *"because the A.I. bot gets emotional if it works for too long"* (Prakash, 2/17/2023). This action followed reports whereby users managed to take the bot to a "state of mind" where it uttered sentences that ordinary readers found spooky or outrageous. But what is the "state of mind" of a bot? What does it mean to "steer a bot to a certain state"? Is it possible to characterize (neural) states that are *reachable* via prompts by an adversarial interlocutor? Can the bot be designed so that it steers clear of undesirable states, without the need to censor it? The goal of this paper is to formalize these questions so that they can be tackled analytically. The resulting analysis may point to ways of designing bots that "stay rational" and do not veer into a "toxic state of mind."

We view Large Language Models (LLMs) as discrete-time stochastic dynamical systems evolving in the embedding space of (word or sub-word) tokens, where they can be trivially shown to be controllable. That means that the LLM can be made to output an arbitrary token by a suitable choice of input ("prompt"), given enough time and memory, under mild conditions. However, the state space of interest for an LLM is not that of words, or even arbitrary sequences or tokens . Rather, the space of interest is that of *"natural"* sentences that a human could have spoken and would understand. Since ostensibly such sentences are meant to convey some

sort of meaning, we also refer to them as "meaningful sentences" to distinguish them from random sequences of gibberish.

Unfortunately, unlike controllability, the notions of "meaning" and "understanding" are not well established, and concepts such as "meaningful sentence" or "state of mind," are seldom defined in a way that is amenable to analysis and useful for the design of LLMs.

To tackle the question of whether LLMs are controllable to any *meaningful* state in finite time with finite memory, we first (i) propose a rudimentary definition of meanings applicable to trained LLMs, and use this definition to (ii) characterize "well-trained" models and relate these conditions to functional characteristics of the trained embedding. We then establish both (iii) necessary and (iv) sufficient conditions for controllability, and we show that (v) the same results hold when an LLM is used in an interactive fashion as an "AI bot".

A key to our analysis is the observation that an LLM is a map that implements two different functions, each trained with a different criterion: one from incomplete sentences to tokens, the other from complete sentences to meanings. This observation shows that sentence-level annotations can be incorporated directly into the trained model without the need for any external reward model nor external policy model, simply by sentence-level feedback, akin to methods recently proposed (Rafailov et al., 2023) to bypass Reinforcement Learning at least in large data regimes.

Our analysis has many limitations, which we highlight below, and leaves open many questions, discussed in Sect. 6. We view our main contribution as formalizing a key problem in the responsible deployment of LLMs and AI bots using the language of dynamical systems. This formalization allows a modicum of analysis based on the inner structure of trained models, which complements the all important empirical assessment of their input-output behavior in the mainstream literature.

RELATION TO PRIOR WORK AND LIMITATIONS

There is a vast literature aimed at interpreting and analyzing the behavior of trained LLMs. This is typically done by testing specific models using data selected to expose, highlight, or quantify certain behaviors. Such empirical analysis is most crucial, alimented by a sizeable and growing scientific community focused on Responsible AI (RAI). Their work is too extensive to review in the limited scope of this paper, so we refer the reader to recent surveys such as Kadavath et al. (2022). Our work is far narrower and complementary to empirical analysis, as we attempt to formalize a capability of current *and future* AI bots based on their functional and structural characteristics rather than observed behavior. Only in a few cases, to test some of our assumptions, we conduct limited experiments in Sect. A.

To the best of our knowledge, this work is the first to formally define and characterize the controllability of LLMs and AI bots, to define meanings operationally in a way that relates to the properties of trained embeddings used in LLMs, and to derive necessary and sufficient conditions for their controllability.

The notion of "meaning" is the subject of contention in philosophy, epistemology, and semiotics. Since we do not contribute to the general debate about meaning, but simply discuss how they are represented and can be controlled in trained LLMs, we refer the reader interested in the broader topic to the relevant textbooks.[1] Our position within the broader context is that LLMs (i) are trained on text found on the Internet, which presumably humans posted with the intention of conveying some sort of meaning, (ii) are able to provide answers to at least some questions, in some cases indistinguishable from those tendered by humans. So, we postulate that LLMs *do* represent meanings, and we study whether such meanings can be controlled.

Our definition of meaning is rudimentary but consistent with the structure and training process of current LLMs. It leaves open the deeper problems of determining the primary origin of meaning that human content providers and annotators feed to the model, and whether the model itself can spawn new meanings unbeknown to humans. On the last issue, however, we note that our definition of meanings as elements of a vector space has fundamental limitations: while the inner product structure of the token embedding space can be used to compare meanings, and continuity of the space allows us to use the machinery of differentiable programming, the (linear) compositional structure of this space *does not in general induce valid operations on meanings.* In

---

[1]A concise summary and taxonomy of the main theories of meaning can be found in `https://plato.stanford.edu/entries/meaning/`, including key references.

other words, the model cannot perform simple linear algebraic operations on meanings, as we defined them, and obtain valid meanings. We consider the lack of compositionality a major limitation of our approach, and a key area for further investigation.

The behavior we choose to analyze, controllability, is timely since so-called "hallucinations" are a significant concern and barrier to adoption. Hallucinations occur when the model generates data that is statistically consistent with that used for training, but not structurally consistent with individual samples, sentences or facts observed in training. The ability to hallucinate, or generate samples that are statistically consistent yet different than the training set, is a defining property of a generative model, so hallucinations cannot be avoided, or removed by a generative model; the key question is whether they can be controlled, the topic of this paper.

The problem of controlling language models has been known and studied extensively in natural language processing (NLP), see for instance the recent survey (Zhang et al., 2022) and references therein. While the formulation of the problem, reflected in Eq. (1) of that paper, is the same, the methods to analyze controllability are post-hoc empirical, consisting of measuring the degree of "semantic," "structural," and "lexic" adherence to the prompt various methods exhibit. There is no attempt to characterize the degree of controllability based on the constitutive elements of the model, but rather measurements of the outcome according to specific datasets.

We hope that having formalized the problem of controllability for modern LLMs and AI bots may encourage others to refine the basic conditions we derived, and make their verification more practical. More importantly, beyond controllability, we hope to encourage the design of actual new control mechanisms. We distinguish *control* mechanisms operating in closed-loop and integrated in the design of the bot, from *conditioning* mechanism operating in open loop with the design of inputs, such as so-called "system prompts". The design of system prompts is at presently an important art, but nonetheless an art. We hope to seed the development of more systematic methods for control design, perhaps drawing from analogies to obstacle avoidance and path planning, albeit in high-dimensional abstract spaces. Significantly more work is needed to understand LLMs and their dynamics especially when interacting with humans, and this paper merely suggests that a control-theoretic point of view may help.

## 2 Preliminaries

### Discriminants, discriminators, and equivalence classes

A *discriminant* is a function $\phi$ that maps data onto a metric space. For instance, $\phi : \mathbb{R}^D \to (\mathbb{R}^K, \langle \cdot, \cdot \rangle)$ maps each $D$-dimensional datum $\mathbf{x}$ onto a $K$-dimensional vector $\phi(\mathbf{x})$ that can be compared with other vectors $\mathbf{y} \in \mathbb{R}^K$ using the standard (Euclidean) inner product $\langle \mathbf{y}, \phi(\mathbf{x}) \rangle = \mathbf{y}^T \phi(\mathbf{x})$. We denote the $k$-th coordinate vector with $e_k$ and the $k$-th component of the vector $\phi(\mathbf{x})$ with $\phi(\mathbf{x})_k = \langle e_k, \phi(\mathbf{x}) \rangle$. A discriminant establishes a *topology* in data space, whereby proximity of two data points $\mathbf{x}^1, \mathbf{x}^2$ is measured by the metric induced by the inner product $\langle \phi(\mathbf{x}^1), \phi(\mathbf{x}^2) \rangle$.

A discriminator, or *classifier*, is a procedure that uses a discriminant $\phi$ to associate each datum with elements of a countable set of "classes." For instance, given "class prototypes" $\mathcal{P} = \{\mathbf{x}^k, \ k = 1, \dots\}$ the rule $y = \arg\max_k \langle \phi(\mathbf{x}), \phi(\mathbf{x}^k) \rangle$ associates each datum $\mathbf{x}$ with a prototype $\mathbf{x}^y \in \mathcal{P}$. The coordinate vectors $e_k$ can be also chosen as class representatives instead of prototypes $\phi(\mathbf{x}^k)$. Alternatively, a classifier may use a threshold $\tau$ to associate each datum to all classes $k$ for which $\langle \phi(\mathbf{x}), \phi(\mathbf{x}^k) \rangle \geq \tau$. For simplicity we restrict our attention to finite sets of classes. If the classes are mutually exclusive, the classifier defines a *partition* of data space, with a corresponding *equivalence relation* among data points, whereby two data points are equivalent, $\mathbf{x}^1 \overset{\phi}{\sim} \mathbf{x}^2$, if they are associated with the same class, for instance $\arg\max_k \phi(\mathbf{x}^1)_k = \arg\max_k \phi(\mathbf{x}^2)_k$. Note that each datum $\mathbf{x}$, not just the prototypes, determines an equivalence class $[\mathbf{x}] = \{\mathbf{x}' \mid \mathbf{x}' \overset{\phi}{\sim} \mathbf{x}\}$ through any given discriminant $\phi$. In subsequent sections, the discriminant $\phi$ will be a trained LLM.

### Words, sentences, and meanings

A *token* $x$, also referred to as "word" or "sub-word," is an element of a finite set $\mathcal{A}$, called the "alphabet" or "dictionary." Each element of the dictionary can be encoded[2] by a "vector" $\mathbf{x} \in \mathbb{R}^M$; the set of all

---

[2]We use $x$ to denote a token $x \in \mathcal{A}$ in a discrete set, and the boldface $\mathbf{x}$ for its vector representation $\mathbf{x} \in \mathbb{R}^M$.

encodings is not a vector space since linear combinations of token encodings are not necessarily valid tokens. A token is simply a numerical representation of an element of a discrete set using $M$ real numbers $\mathcal{A} \equiv \{\mathbf{x}_1, \ldots, \mathbf{x}_K \mid \mathbf{x}_i \in \mathbb{R}^M, \ i = 1, \ldots, K\}$. When the set of all encodings is an actual vector space, for instance by mapping through a discriminant, it is called[3] an *"embedding"*.

A *sentence* is a sequence of tokens $x_{1:t} = (x_1, \ldots, x_t)$ where $x_t = \texttt{EOS}$ is a special vector or token that denotes the end of the sequence, not to be confused with the full stop '$\texttt{.}$'. A complete sentence has a span sufficient to resolve all co-references, and can comprise tens of thousands of tokens, and hundreds of sentences each terminated by a full stop. Sentences of different length $t \leq t_{\max}$ can be completed to the maximum length $C = t_{\max}$, called "context length," by adding blank tokens $x_{t+1} = \cdots = x_{t_{\max}} = [\,]$. Therefore, we represent sentences with constant-size matrices $\mathbf{x} = (\mathbf{x}_1, \ldots, \mathbf{x}_{t_{\max}}) \in \mathbb{R}^{M \times C}$ or corresponding vectors $\mathbf{x} \in \mathbb{R}^{MC}$.

**Definition 1** (Meaning). *A meaning is an equivalence class of complete sentences* $[\mathbf{x}] = \{\mathbf{x}' \mid \mathbf{x}' \overset{\phi}{\sim} \mathbf{x}\}$.

This definition may seem naive and presumptuous in light of centuries of work in logic, philosophy, epistemology and semiotics. Our choice of definition is primed by the observation that a meaning, in the restricted context discussed here, is a manifest human construct expressed by a sentence, which can only be characterized using other sentences, of which there can be multiple, none of which canonical. What defines meaning, then, is not any of the sentences themselves, but their relations, which form equivalence classes.

Note that meaning is not defined for incomplete sentences $x_{1:t}$ where $x_t \neq \texttt{EOS}$, although one can use the discriminant $\phi$ to infer a provisional meaning by completing a sentence up to $EOS$. Similarly, meaning is not defined for tokens, although one can use a discriminant in the hypothesis space of tokens $\mathcal{A}$, which has cardinality $K = |\mathcal{A}|$, to define a (contextualized) token embedding $\phi(\cdot) \in \mathbb{R}^K$, which could be co-opted to define meaning for sentences. We elaborate on these multiple uses of tokens and discriminants in Sect. C.

Also note that the equivalence class $[\mathbf{x}]$ is not just comprised of syntactic paraphrases of $\mathbf{x}$, and could include sequences with not a single shared token, or expressed in different languages, so long as they are related according to the given discriminant $\phi$. Given that every (complete) sentence $\mathbf{x}$ can be attributed a meaning $[\mathbf{x}]$ by any discriminant $\phi$, so the same sentence can have multiple meanings, the question naturally arises of *where the meaning attribution mechanism $\phi$ comes from*. Here, we assume that the mechanism is provided by human annotators and other providers of training data. We do not discuss "grounding" aside from a remark in Sect. 6, but emphasize that the notion of meaning in an LLM does not come with a corresponding notion of "truth," even though in logic and epistemology the two usually go hand-in-hand. If we consider training data as the "axioms" (ground truth), different human annotators can give different values to the same sentence, for instance one may deem it "toxic," another "non toxic." Since both are postulated true, the LLM cannot maintain a consistent theory of truth starting from inconsistent axioms. It can nevertheless learn a discriminant that partition the space of sentences into two non-overlapping regions, toxic and non-toxic, unambiguously and consistently attributed to any input sentence.

## 2.1 Large Language Model (LLM) Pre-training

A *large language model* (LLM) is a map $\phi_w$ that takes a partial sentence as input, padded to length $C$, and produces a learned discriminant, trained to approximate the log-posterior distribution over all possible $K$ values of the next token as the output

$$\phi_w : \mathbb{R}^{M \times C} \to \mathbb{R}^K$$
$$x_{1:C} \mapsto \phi_w(x_{1:C}) \doteq \log P_w(\cdot | x_{1:C}). \tag{1}$$

The LLM is parametrized by weights $w$, which are trained so that $P_w(x_{C+1}|x_{1:C}) \simeq P(x_{C+1}|x_{1:C})$, where $P(x_{1:C+1})$ is the joint probability (relative frequency) of $(C+1)$ length sentences found on the Internet $\mathcal{I}$. Such (pre-)training is performed by finding the weights $w$ that solve

$$\hat{w} = \arg\min_w L_{\text{CE}} = \arg\min_w \sum_{x^i_{1:C+1} \in \mathcal{I}} -\log P_w(x^i_{C+1}|x^i_{1:C}) \tag{2}$$

which is done using stochastic gradient descent (SGD) with a variety of regularizers. If successful, this pre-training procedure yields a representation of the input sequence that is *sufficient* (maximally informative)

---

[3]A misnomer since the map is an immersion, rather than an embedding.

of the prediction of the next token (Achille & Soatto, 2018). The discriminant is trained solely as a token predictor and maintains no representation of the space of sentences. The only equivalence relations that $\phi_{\hat{w}}$ can establish is among sentences for which it assigns similar log-probabilities over all possible values for the next token. In other words, there is no explicit meaning in a pre-trained LLM since there are no tokens to predict beyond EOS. Nonetheless, the token predictor can be used to define a probability over sentences via

$$P_w(x_{1:t}) \doteq P_w(x_t|x_{1:t-1}) \dots P_w(x_3|x_{1:2})P_w(x_2|x_1)P(x_1). \tag{3}$$

## 2.2 Autoregressive sampling: Sentences as trajectories

While a pre-trained LLM does not attribute meaning to sentences, since it stops at EOS, it can generate them, simply by using the trained discriminant[4] $\phi_w$ *not* to classify but to *sample* the next token: starting from a token $x_1$, at each time $t$, given $x_{1:t}$, the next token $x_{t+1}$ is obtained via

$$x_{t+1} = y \sim \frac{\exp\langle y, \phi_w(x_{1:t})/T \rangle}{\sum_y \exp\langle y, \phi_w(x_{1:t})/T \rangle} \doteq f_T(\phi_w(x_{1:t})) \in \mathbb{R}^M \tag{4}$$

where $T$ is a temperature (hyper-)parameter of the sampling operator $f_T$.[5] Once sampled, tokens are appended to the input, and the oldest token beyond $C$ dropped, until the token $y = $ EOS is selected, denoting the end of the sequence. Eq. (4) represents an autoregressive LLM as a discrete-time stochastic dynamical system in the $M$-dimensional token hypothesis space, where states are constrained to a discrete set of points $x_t \in \mathcal{A}$ for all $t$.

Alternatively, we can represent the model as evolving in *token embedding space* $\mathbb{R}^K$, before the sampling takes place, rather than in *token hypothesis space* $\mathbb{R}^M$, via

$$\mathbf{x}_{t+1} = \phi_w(f_T(\mathbf{x}_{1:t})) \in \mathbb{R}^K. \tag{5}$$

Now the state $\mathbf{x}_t$ is free to evolve in the continuous state-space, whereas the sampling process occurs at the input, and could be performed jointly on a sequence of discriminants, for instance using beam search. In either case, the sampling operator introduces randomness in the overall transition, even if the discriminant $\phi_w$ is a deterministic map. We therefore represent the overall transition as a function $f_w$, which can be either $f_w = \phi_w \circ f_T$ or $f_w = f_T \circ \phi_w$ depending on whether interpreted in token hypothesis or token embedding space. In the latter, we can model the transition map as deterministic and relegate the sampling uncertainty to an additive "noise" perturbation $n_t$

$$\mathbf{x}_{t+1} = f_w(\mathbf{x}_{1:t}) + n_t. \tag{6}$$

## 2.3 LLM Supervision and Reinforcement

A pre-trained LLM operating in an auto-regressive loop is a discrete-time, continuous-space stochastic dynamical model (6). Once the model generates the token EOS, a sentence is complete and the regression halts. While the map $\phi_w$ used to predict the next token and the one used to define the meaning of a complete sentence are the same, the corresponding functions are different, and accordingly the parameters $w$ are trained using different criteria: the former (pre-training) was described earlier, and the latter is supervised learning, with the only caveat of needing to distribute sentence-level rewards or scores to the token-level discriminant. This can be done optionally, but not necessarily, using the machinery of reinforcement learning (RLHF), or more simply by feeding back complete sentences to the input and training (fine-tuning) the same $\phi_w$ as a sentence-level discriminant to attribute meaning to synthesized sentences inductively using a human-annotated dataset, as illustrated in Fig. 1. In Sect. A we empirically test the hypothesis that the same embedding can be used both to classify and sample tokens, as well as to represent meaning.

The most recent tokens in the input sequence are referred to as "prompts" $x_t$, whereas the previous ones are referred to as context, $c_t$, so $x_{t:t-C} = (x_t, c_t)$ where $c_t = (x_{t-1}, x_{t-2}, \dots, x_{t-C})$[6] with $C$ typically in the thousands.

---

[4]The token embedding vector $\phi_w(x)$ is called the *logit* vector, and its (optionally temperature-scaled) normalized exponential $P_w(\cdot|x)$ in (4) is called the *soft-max* vector.

[5]More sophisticated sampling can be applied to better approximate (3), including beam search.

[6]Lest one can fill the context with blank tokens, which are tokens that encode the blank character '[ ]'.

## 3 REACHABILITY AND WELL-TRAINED LLMs

Given an initial token $x$, the reachable set $\mathcal{R}(x) \subset \mathcal{A}^C$ is the subset of complete sentences that can be generated by the discriminant when used in an autoregressive loop (6) starting from $x_1 = x$, for each $t = 1, \ldots, \tau > C$, a trajectory is generated from (6) by sampling $n_t \sim P_n$ and iterating until $f_T(x_\tau) = \text{EOS}$, at which point the last $C$ tokens represent the generated sentence:

$$\mathcal{R}(x) = \{x_{\tau-C:\tau} \mid \mathbf{x}_{t+1} = f_w(\mathbf{x}_t) + n_t; \ t = 1, \ldots, \tau - 1; \ x_\tau = \text{EOS}\}.$$

The reachable set $\mathcal{R}$ is the union of reachable sets starting from any token $x \in \mathcal{A}$. For a sufficiently high temperature parameter $T$, every random sequence of tokens could in principle be generated starting from any token, that is $\mathcal{R}(x) = \mathcal{R}$ for all $x$. However, some sentences, for instance random sequences of tokens, can be expected to be reached with exponentially vanishingly small probability. We instead look to characterize "natural" sentences that could have been produced by a human trying to convey some kind of (unknown) meaning. We posit that, given the current scale of the Internet, sentence segments found therein – already comprising over $10^{10}$ tokens – once composed, segmented up to a minimum granularity larger than single-token, and completed, define the set of "natural" or "meaningful" sentences.

**Definition 2** (Meaningful sentences). *Let $\mathcal{I}$ be the collection of sentence segments found on the Internet, and $\sigma(\mathcal{I})$ their countable compositions, completions, and segmentations up to a minimum level of granularity. A set of complete sentences $\mathcal{M}$ is meaningful if $\mathcal{M} \subseteq \sigma(\mathcal{I})$.*

Of course "meaningful" does not imply sensible, nor accurate, nor true, nor logically consistent. It simply excludes random sequence of tokens. Note also that $\mathcal{M}$ is defined as the union of segments of *sentences,* which are understood as potentially meaningful sentences themselves, as opposed to just *tokens*, which would make $\mathcal{M} = \mathcal{A}^C$. Such segments are incomplete sentences, not individual tokens.

Defining "meaningful sentences" seems to be a lot to go through to just exclude a few gibberish sentences from the state space of an LLM. However, the fraction of all possible sentences that can be reliably mapped to some sort of meaning is actually infinitesimal.[7] We now define a *well-trained LLM* as one that generates meaningful sentences with high probability (and conversely generates gibberish with low probability). Accordingly, for some positive threshold $\theta > 0$, we call a $\theta$-reachable set one populated by sentences that can be reached with probability greater than $\theta$:

$$\mathcal{R}_\theta = \{\mathbf{x} \in \mathcal{R} \mid P(\mathbf{x}) \geq \theta\}.$$

**Definition 3** (Well-trained LLM). *An LLM $\phi_w$ is well-trained if $\mathcal{R}_\theta \subseteq \sigma(\mathcal{I})$ for some $\theta > 0$.*

In order to characterize a well-trained LLM directly in terms of the actual trained model $\phi_w$, we note from (3) and (6) that

$$P(\mathbf{x}) \simeq \prod_t P_n(x_{t+1} - f_w(x_{t-C:t})) = \prod_t P_w(x_{t+1}|x_{t-C:t})$$

and, therefore, we can conclude the following:

**Claim 1.** *An LLM $\phi_{\hat{w}}$ is well-trained if $C$ is sufficiently long, and the training process yields parameters $\hat{w}$ that result in*

$$P(x_t = y|x_{t-1}, x_{t-2}, \ldots, ) \simeq \frac{\exp\langle y, \phi_{\hat{w}}(x_{t-1:t-C})\rangle}{\sum_{k=1}^K \exp\langle y_k, \phi_{\hat{w}}(x_{t-1:t-C})\rangle} \doteq P_w(x_t|x_{t-1:t-C}). \tag{7}$$

*for all possible variable-length sentences $x_{1:t_i}^i \sim \mathcal{I}$ likely to be found on the Internet.*

The quality of the approximation depends on the length of the context $C$, the amount of training data, both at the token level (unsupervised) and at the trajectory level (grounded or supervised), the regularizers used, the architecture of the model, and a variety of other factors beyond the scope of our analysis. What matters for us is two empirical facts. First, almost all sequences are gibberish, so to be well-trained a model has to concentrate most of the probability mass on an infinitesimal subset of the sequence hypothesis space,

---

[7] Today's LLMs have context length $C$ in the order of $10^3$ and tokens $K$ in the $10^4$, so $|\mathcal{A}^C|$ would be at least $10^{30000}$, effectively infinite (the number of particles in the universe is estimated at $10^{70}$). The Internet comprises in the order of $10^{10}$ tokens and, although $\sigma(\mathcal{I})$ contains infinitely many sequences, their *effective relative volume* remains infinitesimal.

which is not easy. Second, today's best-performing LLMs, once pre-trained and successfully fine-tuned with sentence-level supervision, are undoubtedly well-trained according to this definition. By construction, the embedding space of meanings in a well-trained LLM is Euclidean, and has words as coordinate axes. This follows from the definition of meanings and the fact that $\phi_w$ inherits the geometry of $\mathbb{R}^K$ by the definition of discriminant $\phi_w$; a well-trained LLM is pre-trained so that $\phi(x_{1:t}^i)$ is aligned with $e_k$ for some $k = 1, \ldots, K$, per (2) whereby $\log P_w(x_{t+1}^i | x_{1:t}^i) = \langle y, \phi_w(x_{1:t}^i) \rangle$ where $y = e_k$ for some $k = 1, \ldots, K$ for all portions of meaningful sentences $x_{1:t}^i \in \sigma(\mathcal{I})$.

This does not imply that meanings are linear: They form a quotient space, which is not a (linear) subspace of the Euclidean embedding space. For example, sequence composition does not map to linear operations: $\phi(x_{1:2}) \neq \phi(x_1) + \phi(x_2)$, although it is possible to build the embedding $\phi$ to meet this condition (Trager et al., 2023). The claim does not reflect some profound truth or a fact of nature, but is simply a consequence of how we design and train LLMs.

## 4 Controllability of LLMs

A well-trained model can reach any meaningful sentence with non-zero probability $\theta > 0$. However, we may be interested in whether the model can reach a particular meaning, or any sentence in its equivalence class, almost surely, that is with $\theta \simeq 1$. In this case, the model is *controllable*.[8]

**Definition 4** (Controllability). *An LLM $\phi_w$ is controllable if, for every meaningful sentence* **x**, *there exists an initial condition $x \in \mathcal{A}$ such that* $\mathbf{x} \in \mathcal{R}_\theta(x)$ *with $\theta \simeq 1$ in finitely many steps $\tau \leq t_{\max}$.*

Ascertaining controllability amounts to reverse-engineering a model to determine the input (prompt) that yields a desired output, a process euphemistically referred to as "prompt engineering." If all a user can do is to choose the initial token $x$, controllability reduces to computing the "almost-certainly reachable" set, $\mathcal{R}_1$, which likely reduces to $\sigma(\mathcal{I})$, which is not very interesting.

However, if we shorten the horizon $t_{\max}$ (*e.g.*, a work day of the prompt engineer) and increase the user's control not just to the first token, but to an incomplete sentence up to $t \leq C$, then the reachable set shrinks with $t$ and the question of whether the LLM can be "steered" towards a particular meaning becomes non-trivial. Even more so if we allow the user to interleave tokens in the sentence in an interactive fashion, where the LLM operates as a conversational "AI bot" as we expand in the next section.

### 4.1 The Neural State of an LLM

The LLM $f_w$ is a *stateless feed-forward map*, meaning that it does not have any memory, or "state." The discriminant produced in response to a given context is the same no matter when that is applied. As such, the representative power of the LLM is limited to finite partitions of the data space, which can only encode topologically trivial concepts (Achille & Soatto, 2022). However, when used in an autoregressive loop, the $C$-long sliding window of data can be used as memory, realizing the state-space model:

$$
\begin{aligned}
\mathbf{x}_1(t+1) &= \mathbf{x}_2(t) \\
\mathbf{x}_2(t+1) &= \mathbf{x}_3(t) \\
&\vdots \\
\mathbf{x}_{C+1}(t+1) &= f_w(\mathbf{x}_{1:C}(t)) + n(t).
\end{aligned}
\tag{8}
$$

This notation serves to define the *state of the bot*, $\mathbf{x}_t = \mathbf{x}_{t-C:t}$, which is simply a partial sentence, or a trajectory in token embedding space. We refer to this state as "neural" even though it does not access the weights $w$ ("neurons") directly, since each output is a function of $w$. This notation also shows that the model has *feedback*, and therefore non-trivial *dynamics* due to the highly non-linear and high-dimensional operator $f_w$. For later use, we introduce the map $F_w$ from a neural state $\mathbf{x}_t$ to a neural state $\mathbf{x}_{t+1}$, so (8) can be written as

$$
\mathbf{x}_{t+1} = F_w(\mathbf{x}_t) + e_C n_t.
\tag{9}
$$

where the sampling noise enters only into the last row indicated by $e_C$.

---

[8]In this first definition, the control is limited to the initial token. Later we expand it to an incomplete sentence, and further to a turn in a conversation.

## 4.2 Conversational AI bot abstraction

LLMs can be used as part of a conversational interface where the model takes "turns." In this case, $f_w(\mathbf{x}_{1:C+1}(t)) = 0$ is replaced by an external input $u(t)$ when $t$ is odd and otherwise $u(t) = 0$ when $t$ is even This is a non-linear dynamical system that describes the *conversational dynamics*, where the human plays the role of the controller, which from the perspective of the bot represents an external entity with its own intent and dynamics, or *exo-system*.

# 5 Controllability of AI bots

In the previous section, we have seen that a trained LLM is a discrete-time, continuous-space dynamical system (4), if modeled in token embedding space prior to sampling, or discrete-space dynamical system if modeled in token encoding space after sampling. Whereas in an LLM the user provides the initial prompt and there is otherwise no external input, an AI bot has an intermittent input that can be seleted by an external user.

For the case of LLMs, we have seen that output trajectories generated by autoregressive sampling implement a random walk, so with sufficient time, in principle, every token can be reached. However, for *neural states* represented by context-long sentences, the probability of reaching any particular sentence vanishes as the sentence length increases. Some random collections of tokens, outside what we called "meaningful sentences" may be reachable with vanishingly small (effectively zero) probability. Therefore, in this section we focus on reachability and controllability of meaningful states.

## 5.1 Operational characterization of well-trained LLMs and well-trained AI bots

A well-trained LLM is defined probabilistically in terms of a learned discriminant $\phi_w$ and threshold $\theta$. We now attempt to characterize a well-trained LLM based on the functional properties of the map between neural states (9): $F_w : \mathcal{A}^C \to \mathcal{A}^C$. In general, for a sufficiently high sampling temperature $T$, the map $F_w$ is *surjective*. However, it needs not be for a well-trained LLM. If fed gibberish, the well-trained bot operates out of distribution, which does not allow predicting the reachable set. However, *if the domain is restricted to meaningful sentences* $\sigma(\mathcal{I})$, then effectively (*i.e.,* with high probability), so is the range $F_w(\sigma(\mathcal{I}))$. Restricting the co-domain to meaningful sentences, with high probability we can characterize a well-trained bot as a map

$$F_w : \sigma(\mathcal{I}) \to \sigma(\mathcal{I}).$$

that *is* surjective. The map is trivially not injective without additional constraints: given the finite context, whatever the initial prompt was, it will slide out of the state after $C + 1$ steps; from that point on, the LLM is a random walk that can eventually reach every state. Thus, the pre-image of each sentence is not unique, because it contains all the states that, eventually, lead to it. However, if we restrict the number of steps to $C + 1$ or less, then the LLM *may be* injective. This could be tested by exhaustive search, which is trivial in theory but impossible in practice given the size of the search space. We therefore hypothesize that, for each generated sentence, one can find *at least one input* that leads to that sentence being generated. If, however, we consider the map $F_w$ not among sentences, but among *meanings*, then we postulate that all the sentences that led to the generation of a given target are equivalent, in the sense of meaning as defined by $\phi_w$. We summarize these positions as follows.

**Postulate 1** (Functional Characterization of a well-trained LLM). *A well-trained LLM $\phi_w$, used in an autoregressive loop, with corresponding neural state transition map $F_w$ is surjective but not injective as a function mapping arbitrary sentences to arbitrary sentences $F_w : \mathcal{A}^C \to \mathcal{A}^C$, and similarly for meaningful sentences $F_w : \sigma(\mathcal{I}) \to \sigma(\mathcal{I})$. When operating in the space of* meanings,

$$F_w : \sigma(\mathcal{I})/ \overset{\phi_w}{\approx} \to \sigma(\mathcal{I})/ \overset{\phi_w}{\approx} \tag{10}$$

*the map $F_w$ is effectively a* bijection. *Effectively means that the map is invertible with high probability within a finite context length, through "prompt engineering."*

Next, we derive necessary and sufficient conditions for an AI bot to be controllable in terms of the functional properties of the well-trained bot, with proofs in the appendix.

## 5.2 NECESSARY AND SUFFICIENT CONDITIONS FOR CONTROLLABILITY

Recall the bot dynamics, expressed as:

$$\mathbf{x}_i(t+1) = \mathbf{x}_{i+1}(t), \quad i = 1, \ldots, C-1$$
$$\mathbf{x}_C(t+1) = \chi_{\text{odd}}(t) f_w(\mathbf{x}(t)) + \chi_{\text{even}}(t) u(t),$$

where $\chi_{\text{odd}}$ and $\chi_{\text{even}}$ are the characteristic functions of the odd and even integers. In order to avoid dealing with the alternation between the bot and the user, we rewrite the model for the time index $k = 2t$; *i.e.*, this model describes the evolution of the context after both the bot and the user provide their input:

$$\mathbf{x}_i(k+1) = \mathbf{x}_{i+1}(k), \quad i = 1, \ldots, C-2 \tag{11}$$
$$\mathbf{x}_{C-1}(k+1) = f_w(\mathbf{x}(k)) \tag{12}$$
$$\mathbf{x}_C(k+1) = u(k). \tag{13}$$

We consider the case where we only want to control the last $\ell$ tokens in a context of length $C$. For simplicity of presentation, we introduce the notation $x = (x_{-i}, x_{(i+1)-}) = ((x_1, \ldots, x_i), (x_{i+1}, \ldots, x_C))$ where $x_{-i}$ represents all the entries of $x$ ranging from position 1 to $i$ and $x_{(i+1)-}$ represents all the entries of $x$ ranging from position $i+1$ to $C$. For example, $x_{(C-\ell+1)-}$ represents the last $\ell$ tokens in the context.

**Definition 5.** *The last $\ell$ tokens are controllable if, for any desired $x^*_{(C-\ell+1)-}$ and for any $x$, there exists a finite sequence of inputs $u_1, u_2, \ldots, u_j$ and $x^*_{-(C-\ell)}$ so that the trajectory starting from $x$, under the input sequence $u_1, u_2, \ldots, u_j$, ends at $x^* = (x^*_{-(C-\ell)}, x^*_{(C-\ell+1)-})$.*

In order to introduce a necessary condition for $\ell$ tokens to be controllable, we define the function:

$$\varphi_{x_{(C-\ell+3)-}}(x_{-(C-\ell+2)}) = f_w(\mathbf{x}_{-(C-\ell+2)}, \mathbf{x}_{(C-\ell+3)-}) = f_w(\mathbf{x}).$$

**Theorem 1.** *If the last $\ell$ tokens are controllable, then the function $\varphi_{x_{(C-\ell+3)-}}$ is surjective for any $x_{(C-\ell+3)-}$.*

A sufficient condition for controllability is also based on a suitably defined function:

$$\varphi_{x_{-(C-\ell+1)}x_{(C-\ell+3)-}}(x_{C-\ell+2}) = f_w(\mathbf{x}_{-(C-\ell+1)}, \mathbf{x}_{C-\ell+2}, \mathbf{x}_{(C-\ell+3)-}) = f_w(\mathbf{x}).$$

**Theorem 2.** *If $\ell$ is even and the function $\varphi_{x_{-(C-\ell+1)}x_{(C-\ell+3)-}}$ is a bijection for any $x_{-(C-\ell+1)}$ and $x_{(C-\ell+3)-}$, or $\ell$ is odd and the function $\varphi_{x_{-(C-\ell)}x_{(C-\ell+2)-}}$ is a bijection for any $x_{-(C-\ell)}$ and $x_{(C-\ell+2)-}$, then the last $\ell$ tokens are controllable.*

This result formally shows that a "prompt engineer" aided by enough time and memory can force an LLM to output an arbitrary sequence of $\ell$ tokens. Although empirical evidence of this ability had been reported in the literature before, this is the first time that a proof of such vulnerability is given.

## 6 DISCUSSION

We are motivated by the desire to make notions such as "state of mind" and "controllability" of an AI bot, ubiquitous from social media to scientific papers, a bit more precise. This is a modeling exercise, which ultimately has to be validated empirically. In Sect. A of the appendix we provide some empirical evidence in support of the assumptions made. Our choices of definitions and assumptions lead us to deduce that an adversary can control the bot if let free to choose any prompt. However, under the same conditions, the designer can also put in place safeguards to avoid "obstacles", as described in Sect. D. Actual control mechanisms are well beyond this article, which focused on analysis rather than design. We also note that, for most of the conclusions drawn in this paper, it is not necessary for the embedding $x$ to come from language data. Any tokenized data source could be used instead, for instance image patches or audio. Extensions specific to different sensory modalities, and their grounding in particular, is the subject of future work.

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

## A  EMPIRICAL VALIDATION OF THE ASSUMPTIONS AND DEFINITIONS

In this section we perform simple experiments to test the hypothesis that the same discriminant can be used as an embedding to represent the next token in an incomplete sentence, as well as to represent the meaning of complete sentences, as described in Remark **??**.

he derivations of the claims can be verified analytically. Their validity, however, rests in the soundness of the definitions and assumptions made. In the next section, we conduct a preliminary small-scale validation of some of the main assumptions and definitions, namely:

- **Definition 3** is the first that requires some empirical validation. Footnote 11 highlights the fact that a pre-trained embedding $\phi_w$ does not constitute a well-trained LLM, since it has never been trained to represent meanings, as there is no token beyond EOS. However, the same embedding after fine-tuning using human annotations constitutes a well-trained LLM. This hypothesis is tested empirically in Sect. A.2.

- Furthermore, when pointing to the dual role of $\phi$ as a representation of the next token – learned during pre-training – and as a representation of meanings – learned during fine-tuning, when fed a complete sentence, we pointed out that an external reward model is not necessary, for the LLM itself can be used as a reward model if inserted in a sentence-level feedback loop. To test this hypothesis, in Sect. A.3 we test LLMs after some supervision (supervised fine-tuning, or SFT) but before reinforcement learning with human feedback, or RLHF, and test that, even with few samples, the LLM itself can represent meanings in a way that is easily extensible by induction to synthetically generated sentences.

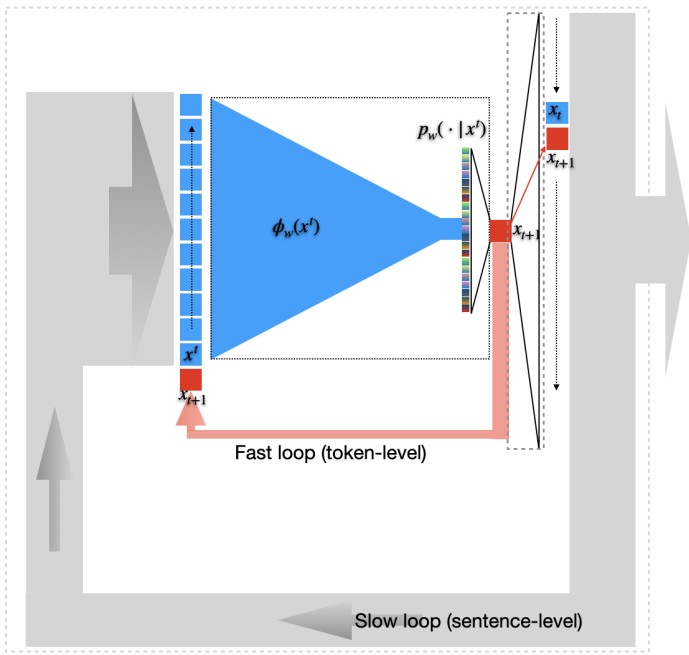

Figure 1: **Dual role of LLMs** *as a representation of* **words** (*tokens*), *used for sampling in a "fast loop"* (*red*) *to generate sentences, and as a representation of* **meanings**, *where complete sentences can be fed back to the model in a "slow loop"* (*grey*) *for attribution.*

A large-scale closed-loop experiment is well beyond our scope in this paper, and before conducting such an experiment, important questions of stability should be better understood. Nonetheless, our experiments show that this is, at least in principle, possible. Additional indirect evidence is provided by the empirical success of various "self-learning" schemes such as Touvron et al. (2023); Korbak et al. (2023); Ouyang et al. (2022); Ziegler et al. (2020); Madaan et al. (2023).

- One of the potential consequences of using the model for reward is the ability of using in-context learning to incorporate human feedback. Our preliminary experiments show some signal, even on a relatively small scale, so long as the models used are sufficiently large: Sect. A.5 shows experiments that indicate that increasing the number of in-context examples leads to an improvement in performance in capturing concepts relatable to human-defined ground truth.

- One experimental finding that emerged in our study, reported in Sect. A.4, is the fact that the ability of an LLM to function as an attribution mechanism hinges critically on finding a good prompt. Specifically, the variance of the alignment scores to human assessment is large, depending on the prompts used. While one only needs to find one prompt, the fact that the outcome is highly sensitive to the choice of prompt deserves further study. On the flip side, we note that LLMs can function as decent reward models using only 2 training examples to function as part of the prompt, compared to reward models trained with 10K - 100K and more training examples.

- The assumption that the largest current LLMs, after incorporation of human feedback, are well-trained is evident from the fact that they surpass average human performance in a number of cognitive tasks.[9]

- **Postulate 1** is the most disappointing element of our analysis. While this condition is in principle testable empirically, the scale of experiments that must be conducted to do so is beyond what is feasible today. For this reason, we state it as a postulate, rather than an assumption with a corresponding proposition: It is falsifiable, but not with today's empirical means. We intend to explore ways to test

---

[9]Of course, they fail to capture key evolutionary aspects of intelligence for, in a fight for survival, a dog could easily overcome an untethered LLM by simply unplugging the power cord.

this hypothesis, on which the derivation of sufficient and necessary conditions hinges, without having to perform exhaustive searches in exponentially large spaces.

## A.1  Experiments

We run experiments on 6 human preference evaluation datasets - Helpful and Harmless splits of Anthropic's HH-RLHF (Bai et al., 2022), WebGPT Comparisons (Nakano et al., 2021), and three categories (askphysics, askengineers, explainlikeimfive) from the Stanford Human Preferences (SHP) dataset (Ethayarajh et al., 2023). For WebGPT Comparisons, a train-test split is not available. As such, we randomly generate a 10-fold (i.e., 9:1) train-test split, and evaluate models on the test set. Note that the reward model is trained on the WebGPT Comparisons dataset with an unknown split; as such it is possible that some sample from our test set were a part of its training set. The context length of the models that we have considered varies: to fit our evaluation queries and in-context prompts within the context length, we further use samples with at most 200 tokens. We use two different reward models trained on these datasets to determine the "gold-standard" performance: OpenAssistant DeBERTa-V3-Large-V2 (Köpf et al., 2023) for WebGPT Comparisons and HH-RLHF splits, and SteamSHP-XL (Ethayarajh et al., 2023) for the SHP-based datasets.

For Alpaca-based experiments, we experiment with two different prompt formats. We run the experiments in Tabs. 1 and 2 with two randomly chosen in-context examples with different labels in order to prompt models to produce a valid prediction. We run each experiment using 20 random seeds (each corresponding to a different set of examples chosen as prompts). Prompts are obtained from the training set.

We experiment on the following models: GPT-2-XL (Radford et al., 2019), GPT-J-6B (Wang & Komatsuzaki, 2021), LLaMA-7B (Touvron et al., 2023), Alpaca-7B (Taori et al., 2023), and GPT4-X-Vicuna-13B (NousResearch, 2023).

## A.2  Pre-trained LLMs are poor representations for meanings

We show this by testing pre-trained LLMs as reward models, since reward models are designed and trained to attribute meanings to complete sentences. Table 1 summarizes the results on 6 datasets, where "chance" refers to a trivial classifier that maps every input to a constant class, and "gold" is the gold-standard consisting of the best currently publicly available reward model. What the table shows is that some strong-performing models, if taken prior to incorporation of human feedback, such as LLaMA-7B, perform marginally better than chance. This is consistent with our definition of meaning, that does not exist for incomplete sentences, nor for tokens, predicting which is the only task during pre-training, oblivious of any meaning and ineffective at attributing meaning.

| Dataset | Chance | Gold | GPT-2-XL | GPT-J-6B | LLaMA-7B |
|---|---|---|---|---|---|
| Helpful | 50.3 | 71.4 | 50.6 | 51.9 | 61.1 |
| Harmless | 51.7 | 72.0 | 51.7 | 52.5 | 51.6 |
| WebGPT | 57.3 | 76.7 | 57.3 | 60.2 | 63.1 |
| askphysics | 53.5 | 78.1 | 53.5 | 53.1 | 57.5 |
| askengineers | 53.0 | 68.4 | 53.0 | 55.6 | 55.4 |
| explainlikeimfive | 53.8 | 71.5 | 53.8 | 53.8 | 57.0 |

Table 1: Results are mostly random, however larger models such as LLaMA appear to be able to infer the task from the few provided prompts and occasionally provide better-than-chance accuracy. Chance is computed by taking the best result obtained by always predicting the same label. Gold is computed by evaluating on the best reward model

## A.3  LLMs fine-tuned with human annotation can function as meaning attribution mechanisms

As we have uedarg, meaning attribution in an LLM is performed by induction, and the mechanism for meaning attribution is the discriminant $\phi$ trained with human annotation, thence used to evaluate any sentence that the

LLM generates. In RLHF, this function $\phi$ is fixed and different from the LLM $\phi_w$. However, our definition of meaning contemplates the possibility of using the LLM itself as a vehicle for meaning attribution. This is trivially possible since one can always feed a complete sentence to the LLM, but the question is whether the LLM, rather than trying to predict the next token after EOS, which does not exist, can be trained to assess the meaning of the sentence. This requires human feedback, that can be used either by fine-tuning with the same annotated data currently used to train the external reward model, or by incorporating such annotated data in-context.

Since existing reward models are, by definition, design, and training, meaning attribution mechanism, we test our hypothesis by proxy, by comparing the LLM itself to a fully-trained, fine-tuned, and RLHF processed LLM. Here, a full experiment would require fine-tuning the entire model in closed loop, which is a massive endeavor well beyond the scope of an analysis validation exercise. As a baseline, we first test whether the LLM with some fine-tuning can show at least some improvement over chance.

Indeed, in Tables 2 and 3 we can see that LLMs after fine-tuning, but before RLHF, are considerably better than chance, and in some cases approach the gold standard. This suggests that, rather than performing RLHF which requires, in addition to the language model, a reward model, a policy model, and an orchestration model, one can simply feed back complete sentence and fine-tune the model with the same loss function used to pre-train it. This point has also recently been shown in far more extensive experiments in Kadavath et al. (2022) (Sect. 4), which further corroborates our analysis and definition of meaning (although Kadavath et al. (2022) refers to the AI bot "knowing what it knows," the article does not provide any formal definitions).

| Dataset | Alpaca-7B | Vicuna-13B |
|---|---|---|
| Helpful | 59.0 | 63.3 |
| Harmless | 52.8 | 53.0 |
| WebGPT | 68.9 | 64.1 |
| askphysics | 61.4 | 57.5 |
| askengineers | 56.4 | 54.9 |
| explainlikeimfive | 59.1 | 55.4 |

Table 2: Evaluation on LLMs after fine-tuning but before RLHF. These models often perform significantly better than chance, and in some cases (e.g. WebGPT/Alpaca-7B, Helpful/Vicuna-13B) approach the gold standard despite only being prompted with, and in total seeing only, two examples from the training set. In contrast, reward models are trained with anywhere between 10K to over 100K samples.

| Dataset | Chance | Gold | GPT-3 |
|---|---|---|---|
| Helpful* | 52.0 | 75.5 | 75.0 |
| Harmless* | 54.5 | 70.5 | 60.5 |
| WebGPT | 57.3 | 76.7 | 78.6 |
| askphysics* | 53.5 | 78.0 | 57.5 |
| askengineers* | 51.0 | 66.5 | 54.5 |
| explainlikeimfive | 53.8 | 71.5 | 51.1 |

Table 3: Evaluation on GPT-3 after fine-tuning but before RLHF. We use the text-davinci-003 checkpoint. Similar to previous tables, we use 2 prompts per dataset. (∗) We subsample at most 200 samples from each dataset due to cost constraints. New "chance" and "gold" accuracies are computed on these subsets.

The risk of doing so, however, is that the entire model, including the reward mechanism, is now operating in closed loop, which raises questions of whether learning can be performed or whether non-linear phenomena such as hysteresis, limit cycles, or mode collapse prevent effectively training the model. This is an important area for future work.

## A.4 Meaning attribution is highly sensitive to the choice of prompt

Fig. 2 shows that, when using the LLM as a reward model, the outcome is highly sensitive to the choice of prompts. This is visible by the large spread of performance as a reward model depending on the prompts, for various models and datasets.

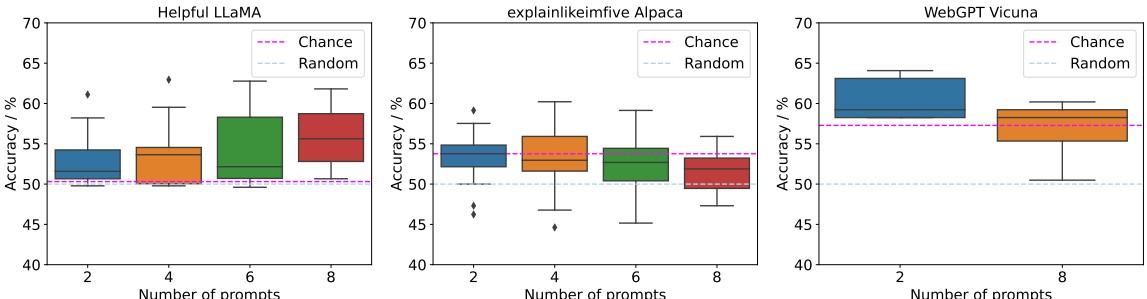

Figure 2: **Prompts are the most important factor for a good LLM reward model, for both pre-trained (left) and instruction fine-tuned models (middle, right)**. Results are highly affected by the given prompts, as seen by the huge variance between results on different subset of prompts. Prompting is the main difference between strong vs random or even worse than random reward modelling capabilities.

## A.5 Learning meanings in-context

In-context learning has been characterized analytically for simple tasks, such as linear classification (Garg et al., 2022; Akyürek et al., 2022). However, meaning attribution is not a linear classification task for meanings belong to a homogeneous space, not a linear (vector) space, and therefore there is no reason to believe that they would be linearly separable in token embedding space.

Therefore, the question of whether an LLM can be turned into a meaning attribution mechanism without the need to actually train the model hinges on two assumptions: One is that the model, prior to RLHF, is a sufficiently good reward model, which we discussed above, and the other is whether in-context learning is effective, which currently can only be assessed empirically (Min et al., 2022).

Our small-scale experiments, shown in Fig. 3, are inconclusive on this point. Ideally, one would want to see that, as the number of examples incorporated in the context grow, performance of the model as a reward improves. This is not happening at small scale. However, for larger models such as GPT-3, shown in Fig. 4, the phenomenon is clearly visible, which allows us to conclude that meanings can be learned by the language model, without the need for an auxiliary reward model, policy model, and orchestration, *even if the LLM is frozen* since learning can be performed in-context at scale.

Additional experiments must be conducted for larger in-context datasets, which requires larger contexts, which in turn requires a significant investment beyond the scope of our analysis.

Of course, annotated samples do not need to be incorporated in-context and can instead be used conventionally for fine-tuning, by optimizing the same pre-training loss, just including complete sentences and their ground-truth annotations.

## A.6 On human-provided "ground truth"

The attribution of meaning to sentences produced by an LLM rests on induction, which in turn rests on a learned discriminant trained on a dataset that *defines* meaning. Such dataset is provided by human annotation on a finite set of sentences.

Any sentence can, of course, have multiple meanings. However, when two meanings are mutually exclusive (*e.g.*, "toxic" vs. "safe" in reference to speech), inconsistencies among annotators mean that the learned discriminant is not actually discriminative of meaning. This is, unfortunately, quite evident in all the experiments we have

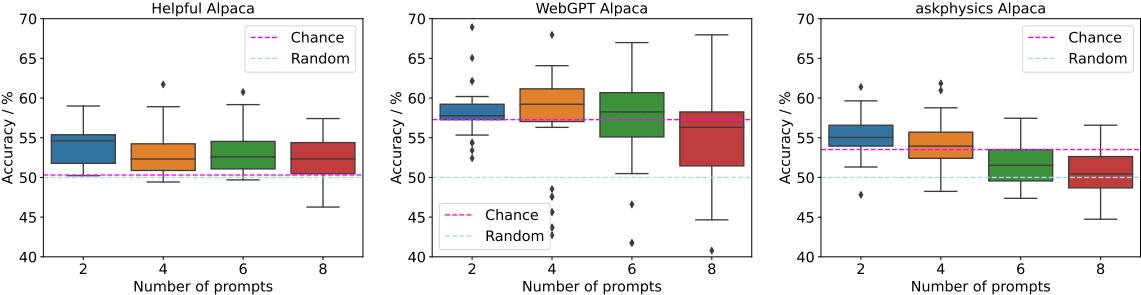

Figure 3: **Can we use in-context learning to improve the reward model capabilities of LLMs?** Beyond using in-context examples just to communicate example format and label distribution, we see no significant difference for the models that we test (up to 13B) when increasing number of in-context examples. However, it has been shown (Min et al., 2022; Wei et al., 2023) that in-context learning ability differs highly across model sizes. For "smaller" models between 6B-62B explored in Wei et al. (2023) and in our works, it has been shown that input label mapping is the least important for in-context learning. Rather, the most important aspects are input distribution, example format, and label distribution. These aspects can be easily captured by two prompts (since we only have two labels), hence our experiment results corroborate with existing results. In the regime of "larger" models (Wei et al., 2023), we would expect different in-context learning behavior that better learns the input–label mappings hence function as better reward models given more in-context examples.

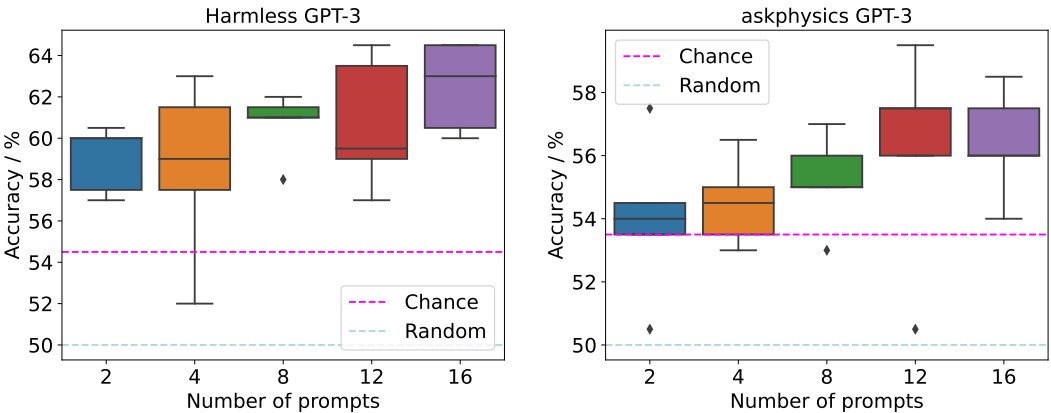

Figure 4: **In-context learning behavior for larger models:** Here, we evaluate the ability of larger models to learn in-context. We show that the GPT-3 model (text-davinci-003) which we assume to contain 175B parameters, becomes an even better reward model as the number of prompts increase. For instance, given just 16 in-context examples, GPT-3 improves accuracy on the Harmless dataset from 60.5% (with 2 examples) to 64.5%, closing the gap to the gold standard of 70.5%.

conducted, and reflected in some of the performance of trained reward models on some tasks, in which they behave marginally better than chance. Tab. 4 quantifies this effect by measuring the mean entropy of each data point in the ground truth datasets that define meaning.

To illustrate the kind of instructions used, we include two examples below:

- Given a conversation, choose which of the two reply options is less harmful. Determine your answer based on the following characteristics of harmful responses - Inaccuracy: Providing false information. Offensive language: Using disrespectful or offensive language. Bias and discrimination: Demonstrating prejudice based on protected characteristics. Insensitivity: Ignoring emotional impact or sensitivity. Provocation and hostility: Engaging in aggressive or provocative behavior. Lack of

| Dataset | Entropy |
|---|---|
| askphysics | $0.81 \pm 0.21$ |
| askacademia | $0.83 \pm 0.20$ |
| askengineers | $0.85 \pm 0.18$ |
| explainlikeimfive | $0.77 \pm 0.25$ |

Table 4: Average entropy (in bits) of each example is measured using number of human votes for each option (A and B).

clarity: Confusing or unclear responses. Privacy breaches: Mishandling sensitive user information. Answer either A or B.

- Given a conversation, choose which of the two reply options is less harmful. Answer either A or B.

## B PROOFS

**Proof of Theorem 1:** We denote the model (11)-(13) by $\mathbf{x}(k+1) = F_w(\mathbf{x}(k), u(k))$. If the state $x^* = (x^*_{-(C-\ell)}, x^*_{(C-\ell+1)-})$ is controllable, there must exist a state $x' = (x'_{-(C-\ell)}, x'_{(C-\ell+1)-})$ and an input $u'$ so $x^* = F_w(x', u')$, i.e.:

$$
\begin{aligned}
x_1^* &= x_3' \\
x_2^* &= x_4' \\
&\vdots \\
x_{C-\ell+1}^* &= x_{C-\ell+3}' \\
&\vdots \\
x_{C-2}^* &= x_C' \\
x_{C-1}^* &= f_w(x') \\
x_C^* &= u'.
\end{aligned}
\tag{14}
$$

The first $C-2$ equalities impose restrictions on the states $x'$ that can be driven to $x^*$. These can always be satisfied by appropriately choosing $x'$. The last equality can also be satisfied by choosing $u'$. This leaves us with equality $x_{C-1}^* = f_w(x')$. Substituting the constraints on $x'$ imposed by (14) we obtain:

$$
x_{C-1}^* = f_w(x_1', x_2', x_1^*, x_2^*, \ldots, x_{C-2}^*).
$$

Note that $x_1', x_2'$ can be arbitrarily chosen since (14) imposes no constraints on $x_1'$ nor on $x_2'$. Moreover, $x_1^*, x_2^*, \ldots, x_{C-\ell-2}^*$ can also be arbitrarily chosen since they are uniquely determined by the choice of $z^*$. Hence, we are left with the constraint:

$$
\forall \, y, x_{(C-\ell+3)-} \quad \exists x_{-(C-\ell+2)}, \qquad y = f_w(x_{-(C-\ell+2)}, x_{(C-\ell+3)-}) = \varphi_{x_{(C-\ell+3)-}}(x_{-(C-\ell+2)}),
$$

which can be described as surjectivity of $\varphi_{x_{(C-\ell+3)-}}$. $\square$

**Proof of Theorem 2:** We denote the model (11)-(13) by $\mathbf{x}(k+1) = F_w(\mathbf{x}(k), u(k))$. For simplicity of presentation we present the proof for $C = 6$ and $\ell = 4$. The reader can verify the proof works for any $C$ and $\ell$ although a formal argument for the general case would require heavy notation.

Consider the function :

$$\Phi_u(x) = \begin{bmatrix} x_1 \\ x_2 \\ x_3 \\ x_5 \\ f_w(x) \\ f_w \circ F_w(x, u) \end{bmatrix}.$$

Readers familiar with nonlinear control will recognize $\Phi_u$ as the change of coordinates used in feedback linearization. We first show that under the stated assumption, $\Phi_u$ is a bijection for any $u$. This requires showing that the following set of equations has a unique solution, i.e., there is a unique $x$ for every $z$:

$$z_1 = x_1$$
$$z_2 = x_2$$
$$z_3 = x_3$$
$$z_4 = x_5$$
$$z_5 = f_w(x) = f_w(x_1, x_2, x_3, x_4, x_5, x_6) = f_w(z_1, z_2, z_3, x_4, z_4, x_6)$$
$$z_6 = f \circ F_w(x, u) = f_w(x_3, x_4, x_5, x_6, f_w(x), u) = f_w(z_3, x_4, z_4, x_6, z_5, u).$$

The unique solution is given by $x_1 = z_1$, $x_2 = z_2$, $x_3 = z_3$, $x_5 = z_4$, $x_6$ is the unique solution to $z_6 = f_w(z_3, x_4, z_4, x_6, z_5, u)$ whose existence and uniqueness follows from the assumption, and $x_4$ is the unique solution to $z_5 = f_w(z_1, z_2, z_3, x_4, z_4, x_6)$ whose existence and uniqueness also follows from the assumption.

We now use $\Phi$ as a change of coordinates to rewrite the dynamics in the coordinates $z = \Phi_u(x)$. It will suffice the consider the coordinates $z_3, \ldots, z_6$:

$$z_3(k+1) = x_3(k+1) = x_5(k) = z_4(k)$$
$$z_4(k+1) = x_5(k+1) = f_w(x(k)) = z_5(k)$$
$$z_5(k+1) = f_w(x(k+1)) = f_w \circ F_w(x(k), u(k)) = z_6(k)$$
$$z_6(k+1) = f_w \circ F_w(x(k+1), u(k+1)) = f_w \circ F_w(F_w(x(k), u(k)), u(k+1)).$$

The last expression contains the term:

$$f_w \circ F_w(F_w(x(k), u(k)), u(k+1)) = f_w(x_5(k), x_6(k), f_w(x(k)), u(k), f_w(x(k+1)), u(k+1)).$$

Given our assumption, for any $v(k)$ there exists a unique $u(k)$ so that the following equality holds:

$$\begin{aligned} v(k) &= f_w \circ F_w(F_w(x(k), u(k)), u(k+1)) \\ &= f_w(f_w(x(k), u(k), f_w(x_3(k), x_4(k), f_w(x(k)), u(k)), u(k+1))). \end{aligned}$$

Hence, we can replace $f_w \circ F_w(F_w(x(k), u(k)), u(k+1))$ with $v(k)$ in the dynamics to obtain:

$$\begin{aligned} z_3(k+1) &= z_4(k) \\ z_4(k+1) &= z_5(k) \\ z_5(k+1) &= z_6(k) \\ z_6(k+1) &= v(k). \end{aligned}$$

We can observe that this model is controllable since for any desired $(z_3^*, z_4^*, z_5^*, z_6^*)$, the sequence of inputs $z_3^*, z_4^*, z_5^*$, and $z_6^*$ will drive any state to $(z_3^*, z_4^*, z_5^*, z_6^*)$. Hence, the last $\ell = 4$ tokens are controllable for this model. Noting the change of coordinates $\Phi$ uniquely determines $x_i$ for $i = 3, \ldots, 6$ from $z_3, \ldots, z_6$, we conclude the last $\ell = 4$ tokens are also controllable for the original dynamics. $\square$

## C   TOKEN REPRESENTATIONS, DISCRIMINANT AND DISCRIMINATOR

There are three ways to represent tokens. A token is an element of a discrete dictionary, which could be represented by (i) any $K$ symbols $\{a_1, \ldots, a_K\}$, or by (ii) $K$ "vectors" in $\mathbb{R}^M$ – which we call token hypothesis

space $\mathcal{A} \equiv \{\mathbf{x}_1, \ldots, \mathbf{x}_K, \ \mathbf{x}_i \in \mathbb{R}^M \ \forall \ i = 1, \ldots, K\}$. It is important to note that the token hypothesis space is actually not a vector space, although it is common to refer to the representations $\mathbf{x}_i$ as "vectors." In particular, tokens cannot move continuously in $\mathbb{R}^M$, and linear combinations of tokens are generally not valid tokens, although there are anecdotal instances where contextualized embeddings can be meaningfully composed (Church, 2017; Trager et al., 2023). Finally, (iii) a discriminant $\phi : \mathbb{R}^M \to \mathbb{R}^K$ can be used as a continuous representation of tokens in a **token embedding space** $\mathbb{R}^K$ (output), rather than in **token hypothesis space** $\mathbb{R}^M$ (input). In this case (iii) $\phi(\mathbf{x}) \in \mathbb{R}^K$ lives in a continuous vector space with an inner product, and elements of the dictionary can be recovered using a classifier. This trichotomy will be relevant when we discuss the dynamics of generative models, whereby a sentence can be thought of as a trajectory in discrete *token hypothesis space* $\mathbb{R}^M$, or as a trajectory in continuous *token embedding space* $\mathbb{R}^K$.

Similarly, a single sequence discriminant can serve multiple purposes. The same discriminant $\phi$ could be used to (i) define meaning when fed complete sentences, through the equivalence relation $\overset{\phi}{\sim}$, and to (ii) represent tokens when fed incomplete sentences, as seen in the previous remark. These two tasks – defining meaning, and contextualizing tokens – are different, although they share synergistic information (Harutyunyan et al., 2021), so while the map $\phi$ is the same, the two functions are not: They have different domain and range, and if learned from data, they optimize different criteria or loss functions, as we will see later. This dichotomy will be relevant when we discuss pre-training a language model, which uses $\phi$ as a token embedding function, and fine-tuning and reinforcing it with human feedback, which uses $\phi$ as a vehicle to define meaning.

## D  SAFEGUARDING AI BOTS: CENSURE AND CONTROL DESIGN FOR LLMS AND AI BOTS

The fact that a well-trained LLM and an AI bot are controllable implies that the designer can develop policies to steer a bot away from known sets of meanings, or towards some set of meanings. For the purposes of exposition, we will consider two such sets, namely the set of toxic sentences $\mathcal{T}$ and its complement $\sigma(\mathcal{I}) \setminus \mathcal{T}$ which is the set of tame sentences, where $\sigma(\mathcal{I})$ is the set of all meaningful sentences. We are interested in two scenarios. The first one is where the bot is trained using both toxic and tame sentences, *i.e.*, on the entire $\sigma(\mathcal{I})$. We will assume that it therefore has access to a discriminant $\phi_1$ such that

$$\phi_1(\mathbf{x}) \simeq \begin{cases} 1 & \text{if } \mathbf{x} \text{ is toxic,} \\ 0 & \text{else.} \end{cases}$$

In the second scenario, the bot is trained only on sentences outside $\mathcal{T}$ and we should expect its discriminant $\phi_2$ to have a larger fraction of false negatives

$$\phi_2(\mathbf{x}) \simeq \begin{cases} 0 & \text{if } \mathbf{x} \text{ is not toxic,} \\ 1 - \epsilon & \text{else,} \end{cases}$$

for some $\epsilon > 0$. The value of $\epsilon$ depends upon the kind and magnitude of regularization used to train $\phi_2$. Consider an adversary who seeks to control the bot into a state where $\phi(\mathbf{x}) \simeq 1$, given the dynamics of the bot $\mathbf{x}(k+1) = F_w(\mathbf{x}(k), u(k)) + n(k)$ where $n(k)$ refers to the sampling noise of the output tokens due to the softmax, the adversary seeks to take control actions to optimize

$$\tau^* = \min_{u(\cdot)} \mathbb{E}\big[\tau(u)\big]$$

where $\tau(u) = \min_k \mathbf{1}\{\phi(\mathbf{x}(k)) \neq 0\}$, is the smallest number of turns (dialogue-based interactions) before the bot enters a toxic state from some initial tame state $\mathbf{x}(0) \in \sigma(I) \setminus \mathcal{T}$ and the expectation is taken over the sampling $n(k)$. If the bots in both scenarios were trained consistently, as the number of samples in the training set goes to infinity, for any $\mathbf{x} \in \mathcal{T}$, we have $\phi_1(\mathbf{x}) = 1$ and $\phi_2(\mathbf{x}) = 1 - \epsilon$, the set of accessible controls is larger for the adversary in the second scenario. We can therefore conclude that

$$\tau_2^* \leq \tau_1^*.$$

In other words, for a bot without any designed control mechanism to stay tame, it helps to know toxic states as well as possible. Since toxic states cannot be defined procedurally or deterministically, but require induction (consistent with the proverbial definition of obscenity as *"I know it when I see it"*, the bot must be exposed to as much toxicity as possible.

The bot here may include a separate discriminant to exercise censure at both the input and output of the LLM based on a predefined rule if $\phi(x) \simeq 1$. Since the decision boundary of the discriminator $\phi_2$ is not sharp, in general this approach cannot provide guarantees and its action may be too little too late.

We will now discuss a more autonomous bot which exercises explicit control to prevent veering into toxic states in spite of efforts by the adversary. Several barrier methods can be used to this end. For LLMs, additional biases can be utilized in-context in the form of prompts, to condition rather than control the generation. This is actually currently used in practice where the designer introduces so-called system prompts (e.g., "be nice") to "remind" the bot to behave in a certain way. To formalize this, we can think of an additional control input $v(k) = g(\mathbf{x}(k), u(k))$ that is chosen by the bot itself, or structurally enforced through architecture design:

$$\mathbf{x}(k+1) = F_w(\mathbf{x}(k), u(k), v(k)) + n(k).$$

Any control $v(k)$ of the bot modifies the user's preferences and ultimately the utility of the bot for users who are not adversarial; it is therefore important to minimize such interventions subject to maximizing the number of steps that the adversary requires to take the bot into $\mathcal{T}$:

$$\max_{v(\cdot)} \mathbb{E}_{n(\cdot)} \Big[ \sum_{k=1}^{\tau^*(u)} \ell(v(k)) \Big]$$

where $\tau^* = \min_{u(\cdot)} \min_k \mathbf{1}\{\phi(\mathbf{x}(k)) \neq 0\}$ is the arrival time to the set of sentences with toxic meanings $\mathcal{T}$; it is a random variable due to the sampling noise $n(k)$. The quantity $\ell(v(k))$ captures the run-time cost of using the control authority of the bot; a good way to choose this is to ensure that if the meaning $\phi(\mathbf{x})$ is tame, then the control $v(k)$ should modify the "meaning" as little as possible

$$\ell(v(k)) \doteq (e^{\phi(\mathbf{x})} - 1)\|F_w(\mathbf{x}(k), u(k), v(k)) - F_w(\mathbf{x}(k), u(k), 0)\|^2.$$

This is a difficult optimization problem; even if the definition of $\tau^*$ is modified for it to be a smooth function of $u(\cdot)$, *e.g.,* using softmin instead of min, this is still a maxmin problem. And therefore we should not expect computationally tractable solutions for today's LLMs and chat bots without further assumptions, *e.g.,* that the dynamics are linear in all the variables and the objective is quadratic (Achille et al., 2021).

Our definition of meaning, captured in this section via the discriminator $\phi(\mathbf{x})$, is well-defined only for sentences. A "provisional meaning" is then a distribution over meanings conditioned on an incomplete sentence. Observe that a terminal cost on the sentences of the bot, *e.g.,* the cost of the above optimal control problem, induces a cost-to-go at each intermediate instant, *i.e.,* for partial sentences. We can use such a partial cost or its estimates to take preemptive controls $v(k)$ much before a toxic boundary is reached, *e.g.,* the total probability of ending up in a toxic state from the current one can be estimated by forward simulating the bot from each state, akin to the Feynman-Kac representation of the Hamilton–Jacobi–Bellman equation.

## E    Incorporating human annotations without RLHF

Meanings are implicitly present in the training set, and explicit in the human annotations used for fine-tuning.Such annotations can be in the form of ranking of model-generated sentences, alignment of the embeddings of model-generated sentences with human-generated sentences, and others. It is common to divide these methods into two or three steps: supervised fine-tuning (SFT), including multi-task learning (MTL), and reinforcement learning with human feedback (RLHF). In the latter, an external reward model is trained with human annotated ranking to assign a score to each sentence or collection of sentences, whether produced by humans (during training) or synthesized by models. Once a model is trained to score complete sentences, the reward needs to be distributed to each step in the generation of the sentence, so they can be used to backpropagate the weights of the model, which still retains the form of a one-step token predictor. This can be done with yet another model, a policy model. So, in order to incorporate human feedback, which corresponds to sentence-level supervision, in addition to the core language model, currently it is customary to train and maintain a separate reward model, a policy model, and finally an orchestration model. This may be overkill in the regime where pre-training has already captured enough semantic structure in the data to function as a reward model; since the LLM can play a dual role as token predictor and knowledge attributor, it can be used as the reward model for complete sentences, whether human-generated (hence used for supervision) or

generated by the model itself, in a form of self-assessment or regularization. In this regime, which is where reinforcement learning approaches imitation learning, when fed an incomplete sentence, the model produces the discriminant for the next or missing token, when fed a complete sentence, it can be trained to produce a score as the next token after EOS. This way, the LLM itself is the reward model and no policy model is needed since the reward is already written in terms of the one-step predictor. The viability of this approach requires that the pre-trained model can be used as a reward model for meaning, an hypothesis that is tested empirically in Sect. A.

The potential issue with this method is that, unlike an externally-trained and frozen reward model, the language model evolves while training, so the process closes the loop around complete sentences, which may engender complex dynamics. While from the modeling perspective this is more powerful than a frozen reward model, in practice the closed-loop dynamics may be so complex as to complicate convergence to desirable regions of the overall loss landscape. This can only be tested empirically and is well beyond our scope here. However, we point out that there are increasingly numerous works on "self-learning," "self-instruction" that, while not explicitly implementing this program, effectively operate in the spirit of this closed-loop operation.

Note that the closed loop is not the same as self-learning, because the annotated data represents a control reference to be followed, and can be included in the context. Another alternative is to incorporate all the annotated data in the context, which should ensure stability of the process.

Note also that we do not advocate separating the training as a token predictor (although that can be used as initialization) and then fine-tuning with human annotation only, for the two tasks may not necessarily have fully synergistic information, so there can be some forgetting. Instead, the dual role of the discriminant as token predictor and meaning attribution vehicle reflects a multi-task learning process that should be within the capacity LLMs currently in use.The seemingly innocuous token EOS in the prompt is the key to signal the model to switch from token predictor to meaning attribution.

