# OpenReview forum: "Taming AI Bots: Controllability of Neural States in Large Language Models"
_ICLR.cc/2024/Conference — Submitted to ICLR 2024_

### Official Review · Reviewer_vfen · 2023-10-27

**Soundness:** 3 good
**Presentation:** 3 good
**Contribution:** 2 fair
**Rating:** 5
**Confidence:** 2

**Summary:**

The authors consider the problem of controlling LLMs. In general, the authors seek to find whether an arbitrary well-formed output can be generated from a well-trained LLM.

The authors characterize LLMs as discrete-time dynamical systems, moving in the token or embedding space. On top of this low-level representation space, the authors consider "meaning" spaces based on discriminators.

Overall, given some assumptions about the surjective nature of LLMs within the set of meaningful sentences, the authors conclude that well trained LLMs are controllable.

**Strengths:**

## Originality
This work appears relatively novel, considering the important problem of LLM controllability and bringing insight from dynamical systems literature to characterize controllability.

## Quality
For the most part, this is a very carefully written theory paper that lays out important assumptions and characteristics for notions of controllability.

## Clarity
The paper was largely clear. I tend to favor figures or diagrams, but the first figure (which I found helpful) only appears in the appendix. I confess that I am typically more of an empiricist than a theorist, but I am familiar with LLMs, dynamical systems, and the math presented in the paper, which I was able to follow.

## Significance
I am undecided about the significance of this work for one particular reason: Postulate 1. The authors are considering a very important problem that many people care about - if a paper can establish that real-world LLMs that we use are controllable, that would be an extremely important paper. The problem, in my mind, is that to reach this conclusion, the authors make a very big assumption about well-trained LLMs in Postulate 1. To their credit, the authors note this limitation, both in the main text and Appendix 1. Overall, I fear that the strength of the assumption in Postulate 1 could significantly limit the significance of this work.

**Weaknesses:**

## Postulate 1:
As noted earlier, the biggest weakness of this work is the strong assumption in Postulate 1 about the bijective/invertible nature of the map $F_w$. To me, this assumption almost gives away the controllability conclusion: if there is an invertible map, then it at least intuitively follows very clearly how such a system would be controllable.

## Presentation of results:
I recognize that the primary contribution of this work is theoretical, rather than empirical, in nature. However, the authors do conduct experiments and present results in Appendix A. Even while keeping results in appendices, I would encourage the authors to at least provide a sentence or two in the main paper highlighting high-level trends of the experiment results.

I wish to emphasize again that I traditional study more empirical AI methods and analysis. Thus, if other reviewers find that this work would contribute to the more theory-focused community, I am happy to change my mind.

**Questions:**

1. In Section 3, the authors describe how, "For a sufficiently high temperature parameter $T$..." random sequences can be generated from any start token. This is true for any $T > 0$, right (barring numerical precision errors from very small numbers)? I do not see why one would need a larger value of $T$ for this theoretical claim (although I recognize that in practice, higher $T$ would "spread out" faster to cover more random sequences).

2. As stated earlier, my main concern about this paper is Postulate 1. What evidence or arguments do the authors have supporting this assumption?

---

> ### Author Response · Authors · 2023-11-22
>
> **Q:** *I am undecided about the significance of this work for one particular reason: Postulate 1. The authors are considering a very important problem that many people care about - if a paper can establish that real-world LLMs that we use are controllable, that would be an extremely important paper. The problem, in my mind, is that to reach this conclusion, the authors make a very big assumption about well-trained LLMs in Postulate 1. To their credit, the authors note this limitation, both in the main text and Appendix 1. Overall, I fear that the strength of the assumption in Postulate 1 could significantly limit the significance of this work.*
>
> **A:** Indeed, we acknowledge that Postulate 1 is a limitation of this paper. However, the assumption are actually not that strong: that current models are well-trained according to our definition can be tested by measuring  perplexity, and qualitatively one would be hard-pressed to argue that at least some current LLMs generate sentences that are statistically distinguishable from human-generated training data, which presumably was written to convey some meaning. We could have made the claim seemingly rigorous by enumeration arguments in the style of old theory papers, so the conclusion would be formally proven. However, we are unsatisfied with this kind of argument which is why we state the proposition as a postulate. By doing so, we  we hope to open a discussion, rather than close it, on this topic that we concur is an important one.
>
> **Q:** *I recognize that the primary contribution of this work is theoretical, rather than empirical, in nature. However, the authors do conduct experiments and present results in Appendix A. Even while keeping results in appendices, I would encourage the authors to at least provide a sentence or two in the main paper highlighting high-level trends of the experiment results.*
>
> **A:** Thank you, will do.
>
> **Q:** *I wish to emphasize again that I traditional study more empirical AI methods and analysis. Thus, if other reviewers find that this work would contribute to the more theory-focused community, I am happy to change my mind.*
>
> **A:** We view ours as more of a position paper, since it opens -- rather than closing -- a key question. We view its contribution as framing the problem in a specific technical language, that of control theory,  which has a rich history that may yield insights for further analysis.
>
> **Q:** *In Section 3, the authors describe how, "For a sufficiently high temperature parameter ..." random sequences can be generated from any start token. This is true for any , right (barring numerical precision errors from very small numbers)? I do not see why one would need a larger value of  for this theoretical claim (although I recognize that in practice, higher  would "spread out" faster to cover more random sequences).*
>
> **A:** Yes, we could just say $T>0$. Thanks for pointing that out.
>
> **Q:** *As stated earlier, my main concern about this paper is Postulate 1. What evidence or arguments do the authors have supporting this assumption?*
>
> **A:** The fact that there exist LLMs that are well-trained according to our definition should not be very controversial, as one could be convinced by simply interacting with some of the best models, or measuring their perplexity on any benchmark validation set. There are also quantitative empirical studies establishing that the best models generate sentences that are indistinguishable from human-generated, with high-frequency/probability. As for the rest of the claim, that follows from the complexity of the trained map coupled with the finiteness of the context. Also empirical evidence is the practice of jailbreaking and adversarial prompting, as well as the massive effort that goes into preventing such in the area of Responsible AI.  However, we recognize that even taking all this into account still does not make a proof.  In our view, though, the framing of the problem in the language of controls still would be beneficial to the community.

---

### Official Review · Reviewer_gqJX · 2023-10-31

**Soundness:** 1 poor
**Presentation:** 1 poor
**Contribution:** 1 poor
**Rating:** 1
**Confidence:** 4

**Summary:**

The paper aims to define "meaning" in the context of LLMs, use this definition to characterize well-trained models, and establish the conditions for controllability of LLMs.

**Strengths:**

The paper sets out to formally characterize controllability of LLMs which is an important issue in preventing adversarial attacks on language models and preventing LLMs from producing undesirable content.

**Weaknesses:**

It's unclear what the contribution of the paper is. The related work section does not connect this paper to specific prior work (only citing two survey papers). Then, most of the paper is spent discussing preliminaries and introducing notation and definitions. The first use of the proposed definitions to make a non-trivial claim is in Section 5 with Postulate 1, but this claim is not justified. Theorem 1 in Section 5 seems to follow directly from the proposed definition of controllability. Theorem 2 is presented with no intuition and the proof in the appendix is only for a special case. It's also not explained how Theorem 2 justifies the main conclusion: that "a prompt engineer aided by enough time and memory can force an LLM to output an arbitrary sequence of ℓ tokens."

**Questions:**

See weaknesses section.

---

> ### Author Response · Authors · 2023-11-22
> **Response to Official Review by Reviewer gqJX**
>
> **Q:** *It's unclear what the contribution of the paper is. The related work section does not connect this paper to specific prior work (only citing two survey papers)*
>
> **A:** The contribution is to have framed the question of controllability of LLMs in the language of control systems: Controllability of dynamical models operating in the quotient space of trajectories determined by the model itself. As for prior work, we are unaware of any work that formalized the problem of controllability of LLMs. If the reviewer is, we would appreciate pointers so we can expand the discussion of relevant related work. We are also unaware of anyone having formalized meanings as pre-images of the output of trained LLMs, which is another contribution of our paper.
>
> **Q:** *Then, most of the paper is spent discussing preliminaries and introducing notation and definitions.*
>
> **A:** Indeed. We believe it is important to spend some effort on the definitions given the level of confusion and disagreement surrounding these concepts. For example, some in the community claim that LLMs are "uncontrollable", without giving any definition, and there is often quoted literature claiming that LLMS "cannot in principle represent meanings", based on a definition of meaning in terms of "intentionality" that is, however, left undefined, leading to tautological arguments. So, we think that within the plethora of empirical papers, a few papers focusing on definitions may not be such a bad thing.
>
> **Q:** *The first use of the proposed definitions to make a non-trivial claim is in Section 5 with Postulate 1, but this claim is not justified.*
>
> **A:** If the reviewer means it is not proven, that is indeed the case. We could have "proven" it by arguing that the context is finite so one can always find the prompt by enumeration, but this would not be practical, so we do not frame the claim as a theorem but rather a postulate. If by "justified" the reviewer means that it not a worthwhile effort to try to study the controllability of LLMs analytically, that would be a judgment call we would disagree with, and stress that of course our analysis is not an alternative but a complement to the necessary empirical assessments, which are currently dominating the literature.
>
> **Q:** *Theorem 1 in Section 5 seems to follow directly from the proposed definition of controllability. Theorem 2 is presented with no intuition and the proof in the appendix is only for a special case. It's also not explained how Theorem 2 justifies the main conclusion: that "a prompt engineer aided by enough time and memory can force an LLM to output an arbitrary sequence of ℓ tokens."*
>
> **A:** The notion of controllability requires verifying the existence of a sequence of inputs for the dynamics (11)-(13) which is a non-trivial endeavor. We further note that inputs appear in equation (13) whereas the assumptions in Theorem 1 relate to the function in equation (12).
>
> In the revised version we will allocate more space in the appendix to provide an intuitive description of Theorems 1 and 2.
>
> As we wrote in the proof of Theorem 2, a general proof would require heavy notation thus rendering it even less accessible to a general audience. Anyone versed in nonlinear controls can verify that the arguments applies in general, whereas readers not familiar with it can more easily follow the special case to gain insight.

---

### Official Review · Reviewer_jHbh · 2023-10-31

**Soundness:** 2 fair
**Presentation:** 2 fair
**Contribution:** 3 good
**Rating:** 5
**Confidence:** 4

**Summary:**

The paper hypothesize that the state space of a language model is sentences understandable to human (rather than sheer tokens or sequence of tokens). It then tries to provide a theory for controllability of language systems in the meaning space.

**Strengths:**

I appreciate the effort to formalize controllability of LLMs as a generic question. This is an interesting topic that can spur further research and help predictability and understand the LLM's behaviours in general. However, the presented theory suffers from various core issues.

**Weaknesses:**

- Definition of equivalence seems to be insufficient and lacks important components.

- Definition of meaning seems to defeat the whole purpose of this paper, as it allows for any gibberish/random sequence of tokens to still induce a meaning and possibly a set of many other gibberish sequences to be in their equivalent class.

- How to find a discriminant for meaning is left out as the authors explicitly assume that “the mechanism is provided by human annotators and other providers of training data.” While the authors emphasize in the introduction that such information can be used in the LLM training without external reward model: “This observation shows that sentence-level annotations can be incorporated directly into the trained model without the need for any external reward model nor external policy model, simply by sentence-level feedback,” I do not see the advantage of this approach over using the very same data to train a reward model and use that either during the training (as in RLHF) or as an augmentation (as in Rectification method), the latter indeed provides quite strong theoretical guarantees. If this secondary goal is valid, the paper requires sufficient reasoning for why the presented approach is superior. If not, the presentation requires to change and reflect only the controllability analysis.

**Questions:**

- Last paragraph of introduction: “we hope to encourage the design of actual new control mechanism” --> It looks like the authors are not aware of the [*Rectification* method](https://arxiv.org/abs/2302.14003), which is indeed a formal control paradigm for avoidance any definable “meaning,” as in the terminology of this paper. See the point bellow.

- Let me first emphasize that I am aware of the differences between these two works (no need to tell me they say X while we say Y). My main point is to help you broaden your research and make the current paper be more comprehensive and useful for the readers, also this is aligned with your last paragraph of the introduction: The concept of probabilistic avoidance, which is akin to *stochastic* controllability but in preventing a meaning from being reached is [studied before](https://arxiv.org/abs/2302.14003). Even though the authors of that paper did not call it controllability (they call it security condition which also comes [from an older paper](http://proceedings.mlr.press/v97/fatemi19a/fatemi19a.pdf)), but it is indeed stochastic controllability in the context of avoidance. To use the terminology of this submission, what that work shows mathematically is that the LLM’s output probabilities can be minimally corrected and that will guarantee the avoidance of any prescribed meaning of interest (the one under their study was toxicity, but essentially the same analysis is applicable to any other meaning). Importantly, such results show that a certain optimal value function corresponding to reaching the prescribed meaning (with reward of -1 and no discount) is key at least in principle for the avoidance, which also highlights the sufficiency of RL as the core learning paradigm for closed-loop controlling/steering an LLM.

- Section Preliminaries, first part --> An LLM in its standard form provides a simplex over token space, not an embedding + a metric. How can an LLM be a discriminant?

- Section Preliminaries, second part --> The definition of equivalence is quite confusing. It looks like that equivalence requires both discriminant and a set of classes, no matter how they are defined (does it?). However, the presented definition only uses discrimanent, which seems incomplete. In that case, $x^1 \stackrel{\phi}{\sim} x^2$ is ambiguous.

- Definition 1 --> [related to the point above] without a set of classes (meanings), the definition is incomplete. Indeed, $\phi$ only maps a given sentence to a metric space, that is a vector $\in \mathbb{R}^K$ (plus a given metric), nothing more. One needs to have (i) a set of classes in addition to (ii) a criteria (like argmax in your example) to define equivalence.

- By extension, $[x]$ does not provide a comprehensible set by just using $\phi$. Intuitively, one can ask, given another vector $y$, then $<\phi(x), \phi(y)>$ must be what in order for $y$ to be part of $[x]$?

- Even if the above issue is resolved, the provided definition of meaning lacks any connection to linguistic meanings, as any random sequence of tokens will induce its own $[x]$. This conflicts with the authors initial claim that they would like to distinguish between a meaningful sentence and a random sequence (first line of page 2). Importantly, this is totally separate from the question of what a good $\phi$ is, and it hold from *any* given $\phi$.

- As for selecting $\phi$, the authors say “Here, we assume that the mechanism is provided by human annotators and other providers of training data.” How is this different from an external reward model, as the authors named as a disadvantage of RL-based methods? (Note: the reward is defined for a complete text not a given point at the middle, so these seem to be exactly the same.)

- Eq 1 --> This is confusing. The range of $\phi$ is $\mathbb{R}^K$, where $K$ was the dimension of matric space, yet the output of $\phi$ is a probability distribution (which is a simplex).

---

> ### Author Response · Authors · 2023-11-22
> **Response to Official Review by Reviewer jHbh**
>
> **Q:** *Definition of equivalence seems to be insufficient and lacks important components. Definition of meaning seems to defeat the whole purpose of this paper, as it allows for any gibberish/random sequence of tokens to still induce a meaning and possibly a set of many other gibberish sequences to be in their equivalent class.*
>
> **A:** Any model trained on natural sentences determines a partition of the space of input sentences, including gibberish, indeed. This is why we restrict the model to operate in the quotient space (10), so input gibberish is excluded from the analysis. In theory, one can of course feed gibberish to a model, but that does not have any meaning in our definition since it is outside $\sigma({\cal I})$. In practice, one can easily exclude gibberish sequences by filtering the input, as done to avoid jailbreaking.
>
> **Q:** *How to find a discriminant for meaning is left out as the authors explicitly assume that “the mechanism is provided by human annotators and other providers of training data.”*
>
> **A:** It is described in in multiple passages (although we can make it clearer) that the discriminant is the trained LLM itself: That means that it is found by training, which is based on human annotations and human-generated utterances. For example, starting from the abstract:
> Page 1: *"The LLM maps complete sequences to a vector space [...] that vector space can be coopted to represent meanings during fine-tuning"*
> Page 2: *"an LLM is a map [...] from complete sentences to meanings"*
> Page 4: Note that the notation used for the discriminant, $\phi$ (Definition 1), is on purpose the same used for the (parametrized) logit vector $\phi_w$ of the trained model (1)
> Page 5: *"by feeding back complete sentences to the input [of the model] and training [...] the same $\phi_w$ as a sentence-level discriminant to attribute meaning to synthesized sentences"*
> Page 6: Claim 1 states that a well-trained model $\phi_{\hat w}$ generates meaningful sentences (and therefore meanings).
> Page 8: establishes that the domain of meaningful sentences is the range of the model $F_w$, again emphasizing that meanings are imposed by the trained model.
>
> Having said that, *we agree* that this point could be expressed more clearly at the beginning: the trained model depends on human-generated content (during pre-training) and human annotation (during fine-tuning), so the origin of meanings is latent in the human content providers and annotators, and the mechanism by which they are transferred to the trained model is the training process. We will try to simplify this explanation and place it at the top of the introduction to avoid any confusion.
>
> **Q:** *While the authors emphasize in the introduction that such information can be used in the LLM training without external reward model: “This observation shows that sentence-level annotations can be incorporated directly into the trained model without the need for any external reward model nor external policy model, simply by sentence-level feedback,” I do not see the advantage of this approach over using the very same data to train a reward model and use that either during the training (as in RLHF) or as an augmentation (as in Rectification method), the latter indeed provides quite strong theoretical guarantees. If this secondary goal is valid, the paper requires sufficient reasoning for why the presented approach is superior. If not, the presentation requires to change and reflect only the controllability analysis.*
>
> **A:** We do not claim that forgoing an external reward model is superior, just that it is sufficient. Indeed the reviewer is correct that, if one can afford it, direct augmentation or RLHF with an externally-trained reward model (which then becomes the origin of meanings) is also possible, and possibly better. All we are saying is that an external reward model is not necessary, because one can use the model itself -- as has been established in various alternatives to RLHF, such as Direct Preference Optimization (DPO). We were not aware of the "rectification method", as discussed below, since that appeared after our paper was completed, but we will include it as an alternative way of endowing the trained model with meanings.

---

> ### Author Response · Authors · 2023-11-22
> **Response to Official Review by Reviewer jHbh (continued)**
>
> **Q:** *Last paragraph of introduction: “we hope to encourage the design of actual new control mechanism” --> It looks like the authors are not aware of the Rectification method, which is indeed a formal control paradigm for avoidance any definable “meaning,” as in the terminology of this paper. See the point bellow. Let me first emphasize that I am aware of the differences between these two works (no need to tell me they say X while we say Y). My main point is to help you broaden your research and make the current paper be more comprehensive and useful for the readers, also this is aligned with your last paragraph of the introduction…*
>
> **A:** Thanks for the useful reference and the thorough description of that work, which we were not aware of since it appeared on ArXiv after our work was completed. We will add discussion as it certainly sounds pertinent, thank you for the pointer and the explanation.
>
> **Q:** *Section Preliminaries, first part --> An LLM in its standard form provides a simplex over token space, not an embedding + a metric. How can an LLM be a discriminant?*
>
> **A:** An LLM is trained so that the vector of logits, or the corresponding soft-max, approximates the (log-)posterior of the next token, which is the optimal (Bayesian) discriminant. So, if you view the LLM as a map from an input sequence of discrete tokens to the softmax vector, it is indeed a discriminant, and since it is trained with the (exponential of the) inner product of the embedding with the one-hot encoding of the next token, the inner product (metric) is built into the embedding space by design of the training process.
>
> **Q:** *Section Preliminaries, second part --> The definition of equivalence is quite confusing. It looks like that equivalence requires both discriminant and a set of classes, no matter how they are defined (does it?). However, the presented definition only uses discriminant, which seems incomplete. In that case, x^1 phi ~ x^2 is ambiguous.*
>
> **A:** The definition requires either a discriminant or a set of classes, since one determines the other. If given a discriminant, one can define partitions using simple order relations or rules (e.g. thresholds), and given a set of classes one can define a discriminant as a (piecewise constant) function.

---

> > ### Author Response · Authors · 2023-11-22
> > **Response to Official Review by Reviewer jHbh (continued)**
> >
> > **Q:** *Definition 1 --> [related to the point above] without a set of classes (meanings), the definition is incomplete. Indeed,  only maps a given sentence to a metric space, that is a vector  (plus a given metric), nothing more. One needs to have (i) a set of classes in addition to (ii) a criteria (like argmax in your example) to define equivalence.*
> >
> > **A:** That is technically correct, and we will clarify in the narrative: We leave (ii) open as a design choice, since typically fine-tuning is done with a variety of ranking (with argmax), or alignment (penalizing the inner product directly) or scoring.  In all cases, the decision rule is based on the embedding and the metric: During pre-training, the classes are represented by the one-hot encoding of the next token, and the inner product with the discriminant (logit vector) is explicit in the empirical cross-entropy loss. During fine-tuning, either through ranking or alignment, the same inner product is used either to rank options (usually expressed in context), to align with the soft-max of the corresponding tokens (1, 2, 3, etc.), or the embedding of natural sentences and generated sentences are compared, again with the same inner product. In all cases, a vector plus an inner product (metric) is sufficient to determine a classifier (discriminant vector).
> >
> > **Q:** *By extension,  does not provide a comprehensible set by just using . Intuitively, one can ask, given another vector , then  must be what in order for  to be part of ?*
> >
> > **A:** As a simple example, consider the meaning "toxic". Although the same sentences can be deemed toxic by one annotator and tame by another, in the end for any chosen threshold the classifier will partition the space of input sentences into two disjoint regions. Then, for a given $x$, the equivalence class $[x]$ is any sentence $y$ whose discriminant falls in the same partition as $x$, which can be determined by the inner product $\langle \phi(x), \phi(y) \rangle$, or the corresponding softmax; since in this binary classification, $P_w(y|x) > 0.5$ if $y \in [x]$, and $P_w(y|x) < 0.5$ otherwise. (If it happens to be exactly $0.5$ up to the numerical resolution, then it is ambiguous). One can do the same with respect to an arbitrary number of classes, each represented by any element (prototype) in the class. Then, belonging to one's class is equivalent to a nearest neighbor rule in the learned feature space $\phi_w(X)$ for $X$ the set of all input sequences.
> >
> >
> > **Q:** *Even if the above issue is resolved, the provided definition of meaning lacks any connection to linguistic meanings, as any random sequence of tokens will induce its own . This conflicts with the authors initial claim that they would like to distinguish between a meaningful sentence and a random sequence (first line of page 2). Importantly, this is totally separate from the question of what a good  is, and it hold from any given .*
> >
> > **A:** The apparent conflict is vacated by the fact that random sequences are not in $\sigma({\cal I})$. There are connections with distributional theories in linguistics, since one can convert sets and partitions to conditional probabilities and vice-versa, as others have shown. Our definition in terms of equivalence classes is also implicit in semantic paraphrasing in linguistics.  Since there is no universal notion of meaning that we know of, data collection yields a distribution of expressions, which is captured by a trained model, which can then use it to partition the input space. Different data, and different models (and different individual humans) all maintain different meaning representations. We therefore limit our claims to the system of meanings of a specific trained model.

---

> > > ### Author Response · Authors · 2023-11-22
> > > **Response to Official Review by Reviewer jHbh (continued)**
> > >
> > > **Q:** *As for selecting , the authors say “Here, we assume that the mechanism is provided by human annotators and other providers of training data.” How is this different from an external reward model, as the authors named as a disadvantage of RL-based methods? (Note: the reward is defined for a complete text not a given point at the middle, so these seem to be exactly the same.)*
> > >
> > > **A:** If an external reward model exists, one can use it to induce a meaning structure, in perfect harmony with our framework. The disadvantage is having to rely on an external model that defines meaning, which then prompts the question of how that model acquires meaning in the first place. Since that model is also trained with human utterances and/or human supervision, then even if using an external reward model, the mechanism by which meanings are ultimately transferred to the trained model is still the same that we describe, originating with human annotators and content providers, then transferred to the trained model during the learning process (directly in SFT, indirectly through the external reward model in RLHF). We will clarify this further and add reference to the possibility of using an external reward model if one so desires.
> > >
> > > **Q:** *Eq 1 --> This is confusing. The range of  is , where  was the dimension of matric space, yet the output of  is a probability distribution (which is a simplex).*
> > >
> > > **A:** $K$ is the number of tokens in the dictionary (so, the discriminant has $K$ components), and $M$ is the dimension of the vectorized embedding of the tokens. The metric space is where the logits live, which is a continuous space of dimension equal to the number of hypotheses, or tokens, which is $K$. This should be clear in equations (4), (5), and (6) which are devoted to stress the point that the same entities (elements of the dictionary) are represented differently at different stages of the LLM.

---

### Official Review · Reviewer_ecwe · 2023-11-06

**Soundness:** 3 good
**Presentation:** 2 fair
**Contribution:** 2 fair
**Rating:** 5
**Confidence:** 3

**Summary:**

This paper studies whether an agent can use prompts to steer LLMs to generate any sentences as wanted. A dynamical system perspective is adopted, and LLMs are viewed as discrete-time systems evolving in the embedding space of tokens. The authors first describe that  ''meanings'' in trained LLMs can be viewed as equivalence classes of complete trajectories of tokens. Based on this viewpoint, they end up with a question of determining the controllability of a dynamical system evolving in the quotient space of discrete trajectories induced by the model itself. Several conditions for controllability are then provided.

**Strengths:**

The paper is very original, and presents some unique idea on how to determine the controllability of "well-trained" LLMs. Connecting LLMs with control is definitely interesting. A new theoretical perspective is developed. This paper may also inspire more researchers to think about the fundamental theory of LLMs.

**Weaknesses:**

1.  The characterization of meanings seems quite subjective. The authors mentioned that their characterization of meanings is compatible with the deflationary theories in epistemology. It seems that this explanation itself does not justify why such a characterization is meaningful for studying the controllability of LLMs in general.

2. The authors claim that their conditions are largely met by today’s LLMs. More justifications are needed. This also seems a hand-waving statement. "Largely met" means "not always met"?

3. Many of the equations are quite difficult to understand. I have tried very hard to follow the theoretical arguments in this paper. However, I still feel very confused in the end. I will ask some questions in the "Question" section.

**Questions:**

1. Does Definition 1 assume some sort of underlying classifier such that the equivalent classes can be defined? What is that specific classifier? The authors mentioned "the mechanism is provided by human annotators and other providers of training data." This is very confusing. I don't understand what exactly this phi is.

2. Equation (6) is very confusing. I mean, in Equation (4), y is sampled from the softmax operation. Then all of a sudden, it becomes an additive noise? It seems that the noise n_t depends on x_{1:t}?

3. Regarding Definition 3, an LLM is well-trained if theta is any small positive number?

4. What does Claim 1 mean? Things that happen before C do not matter? Then how large does this C need to be for practical LLMs? I mean, suppose for a task of writing a book, C has to be really large?

5. What does Postulate 1 really mean? How to justify this?

6. How can we justify the assumptions in Theorem 2?

---

> ### Author Response · Authors · 2023-11-22
> **Response to Reviewer ecwe**
>
> **Q:** *The characterization of meanings seems quite subjective.*
>
> **A:** Indeed, our definition of meaning is specific to a discriminant function $\phi(\cdot)$, which in the case of an LLM is implemented by the trained model $\phi_w(\cdot)$. Different models partition the data differently, although models trained on the same or similar data will likely yield similar partitions. We do not believe in a universal notion of meaning, and in any case our scope is limited to how meanings are represented by LLMs.
>
> **Q:** *The authors mentioned that their characterization of meanings is compatible with the deflationary theories in epistemology. It seems that this explanation itself does not justify why such a characterization is meaningful for studying the controllability of LLMs in general.*
>
> **A:** It is not essential that our definition aligns with any particular epistemological theory, but we note that, once mapped to LLMs,  some theories boil down to representing meanings as equivalence classes of expressions, and then differ on their genesis. In our case, such classes are determined by the state of the trained model, which is what we want to control.
>
> **Q:** *The authors claim that their conditions are largely met by today’s LLMs. More justifications are needed. This also seems a hand-waving statement. "Largely met" means "not always met"?*
>
> **A:** Met by some, but not all, trained LLMs. LLMs by design represent a joint distribution on a sequence of tokens, but how well that approximates the empirical distribution of natural sentences is a matter of approximation rather than constraint satisfaction. In this context, by "largely" we mean that perplexity of trained models is low enough that generated sentences are indistinguishable from human generated ones. This may not be true of all models, so we use "largely" to indicate that the conditions are valid only for some models.
>
> **Q:** *Does Definition 1 assume some sort of underlying classifier such that the equivalent classes can be defined? What is that specific classifier? The authors mentioned "the mechanism is provided by human annotators and other providers of training data." This is very confusing. I don't understand what exactly this phi is.*
>
> **A:** In the definition, $\phi$ can be any discriminant vector. In a trained model, $\phi = \phi_w$ is the soft-max vector after supervised fine-tuning (and optionally RLHF). Since that vector is a function of the training data, both in pre-training and fine-tuning, and the training data is provided by human annotators and authors, ultimately the source of meanings is latent in the expressions, ranking, or scoring humans have provided for training.
>
> **Q:** *Equation (6) is very confusing. I mean, in Equation (4), y is sampled from the softmax operation. Then all of a sudden, it becomes an additive noise? It seems that the noise $n_t$ depends on $x_{1:t}$?*
>
> **A:** Indeed the fact that the same token can be represented in different ways can be confusing at first:  Tokens can be represented as elements of a discrete dictionary after sampling, as in (4), or as discriminant vectors before sampling, as in (5). In the latter case, the effect of different selections in the (previous round of) sampling results in a perturbation of the embedding vector of the next token, which is represented in (6) as an additive perturbation since embedding vectors (logits) are  elements of a vector space. In general, it is true that the ``noise'' $n_t$ is dependent on the previous history, $x_{1:t-1}$, but in complex ways that can be modeled statistically.
>
> **Q:** *Regarding Definition 3, an LLM is well-trained if theta is any small positive number?*
>
> **A:** Yes, although different users, or use cases, may have different standards on what constitutes a well-trained model, reflected in the perplexity thresholds that are considered acceptable.

---

> > ### Author Response · Authors · 2023-11-22
> > **Response to Reviewer ecwe (continued)**
> >
> > **Q**: *What does Claim 1 mean? Things that happen before C do not matter? Then how large does this C need to be for practical LLMs? I mean, suppose for a task of writing a book, C has to be really large?*
> >
> > **A:** Claim 1 says that the condition of being a well-trained LLM can be met simply by predicting the next token, so long as the context is large enough and perplexity is sufficiently low. In other words, Claim 1 connects the definition of being well-trained with the criterion used to train LLMs.
> >
> > **Q:** *What does Postulate 1 really mean? How to justify this?*
> >
> > **A:** It means that one can find a prompt that causes the model to utter any meaningful sentence with high probability. It could be stated as a theorem since the context is finite and one could trivially find the prompt by enumeration, but since today's models have tens of thousands of tokens as context, enumeration is not viable in practice. So, we state it as a postulate in hope that others will find better methods -- analytical or empirical -- to prove (or disprove) the claim. It is admittedly a weakness of our analysis; yet framing the problem as controllability of trajectories in a quotient space determined by the dynamical model itself is novel to the best of our knowledge, and may spark interest in communities beyond NLP.
> >
> > **Q:** *How can we justify the assumptions in Theorem 2?*
> >
> > **A:** Using Postulate 1: if one accepts the enumeration argument, they can consider the conditions of Theorem 2 met.

---

### Author Response · Authors · 2023-11-22
**Overall comments**

We are grateful for all the constructive comments and the recognition of the originality of the approach. We are also appreciative of the
 critiques that helped us identify passages that could be articulated more clearly. We are in complete agreement on the highlighted limitations of the method, and share the assessment that many questions are left open by our analysis. Yet we view the formalization of the problem of controllability of AI Bots in the language of dynamical systems useful to expand the set of tools available to analyze LLMs  beyond the NLP community.

We believe that defining some concepts that are often used informally has value, even if one disagrees with our choice of definition. If so, others can propose different definitions. We note that our notion of meaning pertains to fine-tuned models that use explicit supervision, but is undefined before the EOS token. This is sufficient to analyze controllability but less general than definitions that pertain to (distributions of continuations of) incomplete sentences, which can be applied to pre-trained models. But even in that case, the set of sentences that produce the same distribution of continuations form equivalence classes, which is encompassed by our definition.

Below we answer specific questions, and will incorporate suggested changes and clarifications in the updated manuscript.

Thank you!

---

### Meta-Review · Area_Chair_6cFc · 2023-12-11

**Metareview:**

A dynamical systems point of view on LLMs can be illuminating, and the controllability question (can a prompt steer an LLM to any goal sequence) as posed in this paper is an intriguing one.  However, the assumptions made in the paper failed to convince the reviewers of relevance and realism; other concerns were raised about clarity of presentation and formalism of meaning in terms of equivalence classes of token trajectories.

**Justification For Why Not Higher Score:**

The idea of drawing on tools from dynamical systems theory, such as controllability, to ask whether an LLM can be prompted and steered towards a target sequence, is certainly very appealing conceptually. However, the technical presentation, the notion of equivalence classes to formalize "meaning" and realism of assumptions made in the paper, appeared to uniformly cause confusion among the reviewers that was not satisfactorily resolved.

**Justification For Why Not Lower Score:**

N/A

---

### Decision · Program_Chairs · 2024-01-16

Reject